

# Retrieval of absolute $SO_2$ column amounts from scattered-light spectra - Implications for the evaluation of data from automated DOAS Networks.

Peter Lübcke[1], Johannes Lampel[2], Santiago Arellano[3], Nicole Bobrowski[1,6], Florian Dinger[1,2], Bo Galle[3], Gustavo Garzón[4], Silvana Hidalgo[5], Zoraida Chacón Ortiz[4], Leif Vogel[1,*], Simon Warnach[1], and Ulrich Platt[1]

[1]Institute of Environmental Physics, University of Heidelberg, Heidelberg, Germany
[2]Max Planck Institute for Chemistry, Mainz, Germany
[3]Department of Earth and Space Sciences, Chalmers University of Technology, Gothenburg, Sweden
[4]Laboratory Division, Colombian Geological Survey, Cali & Manizales, Colombia
[5]Instituto Geofísico, Escuela Politécnica Nacional, Quito, Ecuador
[6]Institut für Geowissenschaften, Johannes Gutenberg-Universität Mainz, Germany
[*]now at: Basque Centre for Climate Change (BC3), 48008, Bilbao, Spain

*Correspondence to:* P. Lübcke (pluebcke@iup.uni-heidelberg.de)

**Abstract.** Scanning Differential Optical Absorption Spectroscopy (DOAS) networks using scattered solar radiation have become an increasingly important tool for monitoring volcanic sulphur dioxide ($SO_2$) emissions. In order to get absolute column densities (CDs), the DOAS evaluation requires a Fraunhofer Reference Spectrum (FRS) that is free of absorption structures of the trace gas of interest. At volcanoes, this requirement can be formulated in a weaker form, if there is a plume free viewing

direction within the spectra of a scan through the complete sky. In this case, it is possible to use a specific viewing direction (e.g. zenith) as FRS and correcting for possible plume contamination in the FRS by calculating and subtracting an $SO_2$ offset (e.g. the lowest $SO_2$ CD) from all viewing directions of the respective scan. This procedure is followed in the standard evaluations of data from the Network for Observation of Volcanic and Atmospheric Change (NOVAC). While this procedure is very efficient in removing Fraunhofer structures and instrumental effects it has the disadvantage that one can never be sure

that there is no $SO_2$ from the plume in the FRS. Therefore, using a modelled FRS (based on a high-resolution Solar atlas) is of great advantage. We followed this approach and investigated an $SO_2$ retrieval algorithm using a modelled FRS. In this manuscript, we present results from two volcanoes that are monitored by NOVAC stations and which also often show a large volcanic plume. Results from a DOAS $SO_2$ evaluation using a modelled FRS are presented for data from Nevado del Ruiz (Colombia) recorded between January 2010 and June 2012 and from Tungurahua (Ecuador) recorded between January 2009

and December 2011. Instrumental effects were identified with help of a Principal Component Analysis (PCA) of the residual structures of the DOAS evaluation. The $SO_2$ retrieval performed extraordinarily well with an $SO_2$ DOAS retrieval error of $1 - 2 \cdot 10^{16}$ [molecules/cm$^2$]. Compared to a regular evaluation (using a FRS recorded directly before the scan), we found systematic differences of the differential Slant Column Density (dSCD) of only up to $\approx 15\%$ when looking at the variation of the $SO_2$ within one scan. The major advantage of our new retrieval is that it yields absolute $SO_2$ CDs and that it does not require

complicated instrumental calibration in the field, since the method exploits the information available in the measurements.



We compared our new method to an evaluation that is similar to the NOVAC approach, where a spectrum that is recorded directly before the scan is used as an FRS and an $SO_2$ CD offset is subtracted from all retrieved dSCD in the scan to correct for possible $SO_2$ contamination of the FRS. The investigation showed that 21.4% of the scans (containing significant amounts of $SO_2$) at Nevado del Ruiz and 7% of the scans at Tungurahua had a large difference between the two methods (more than a factor of 2). The overall distribution of the $SO_2$ CDs in one scan can in some cases indicate if the plume affects all viewing directions and thus these scans can be discarded for NOVAC emission rate evaluation. However, there are other cases where this is not possible and thus the reported $SO_2$ emission rates would be underestimated. The new method can be used to identify these cases and thus it can be used to improve $SO_2$ emission budgets.

# 1 Introduction

Since the introduction of the Correlation Spectrometer (COSPEC, Moffat and Millan, 1971; Stoiber et al., 1983) measurements of volcanic $SO_2$ emission rates have become an additional tool for volcanologists to study the activity of volcanoes. More recently the availability of miniature spectrometers allowed the wide-spread application of the well known DOAS technique (e.g., Perner and Platt, 1979; Platt and Stutz, 2008) in volcanic environments (e.g., Galle et al., 2003; McGonigle et al., 2005). Automated systems for plume measurements were subsequently developed based on scanning the volcanic plume from different stationary positions, the so-called scanning-DOAS method (Edmonds et al., 2003). Scanning-DOAS instruments are now installed at many volcanoes in order to monitor $SO_2$ emission rates. One of the first installations were scanning DOAS instruments at Montserrat volcano (Edmonds et al., 2003). The Network for Observation of Volcanic and Atmospheric Change (NOVAC, Galle et al., 2010) is at present composed of more than 80 scanning-DOAS instruments at about 30 volcanoes worldwide. Furthermore, Etna and Stromboli, Italy, are both monitored by a comparable scanning-DOAS network (FLAME network, Burton et al., 2009). Another approach using similar instruments is the so-called Hawaiian FlySpec fence-line at Kilauea volcano which consists of ten fixed, upward looking spectrometers (Businger et al., 2015).

In order to correctly retrieve absolute $SO_2$ CDs from the recorded spectra, and thus calculate accurate $SO_2$ emission rates, a background spectrum, which is free of volcanic absorption features, is required. Typically, DOAS $SO_2$ evaluations use a FRS recorded directly prior to the scan (for example with a different viewing direction), to correct for the strong Fraunhofer lines of the solar spectrum. Contamination of this FRS with volcanic $SO_2$ absorption structures can in principle be corrected for by introducing an $SO_2$ CD offset that is subtracted from all $SO_2$ CDs of the respective scan (details about the calculation of this offset is provided in section 2). However, if all viewing directions contain absorption signatures of volcanic $SO_2$ this approach leads to an incorrect offset and thus erroneous $SO_2$ CDs.

Therefore it is desirable to use a universal FRS, which is free of $SO_2$ (or in general free of the trace gas to be measured). First investigations on using an artificial background spectrum as an FRS for the DOAS evaluation of volcanic $SO_2$ were performed by Salerno et al. (2009). Burton and Sawyer (2016) use a similar approach of modelling the background spectrum based on a high-resolution Solar atlas for their iFit method, a direct fitting approach for the evaluation of volcanic $SO_2$ and $BrO$.



This work will follow the idea of using a high resolution Solar atlas spectrum (Chance and Kurucz, 2010) in order to calculate a gas free background spectrum which is used as an FRS for the DOAS evaluation of $SO_2$. First steps towards the here described approach were taken in Lübcke (2014), where measurements from NOVAC instruments at Nevado del Ruiz were evaluated for $SO_2$ with a modelled background spectrum. The manuscript is structured in the following way:

We applied the $SO_2$ retrieval with the modelled FRS to data from NOVAC in order to study how frequently differences between the standard evaluation and the approach using an artificial reference spectrum can be observed. In addition, we investigated the remaining residual structure of the DOAS evaluation with help of a Principal Component Analysis (PCA). The results of the PCA were used in order to take instrumental effects into account and improve the retrieval. The evaluation process and possible pitfalls will be described in detail. We focussed on two large volcanic emission sources for our study:

Spectra were evaluated from two instruments at Nevado del Ruiz (Colombia), covering the time between January 2010 and June 2012, and from 3 instruments at Tungurahua (Ecuador), covering the time between January 2010 and December 2012.

We compare the results of an evaluation using a modelled FRS to a standard NOVAC evaluation, which uses a spectrum measured with an upward-looking viewing direction as FRS. In the standard NOVAC approach, possible $SO_2$ contamination of the FRS is corrected for by subtracting an $SO_2$ offset value. In this manuscript we defined the offset as the lowest $SO_2$

CD of the respective scan. The large NOVAC database gives us the possibility to determine how frequently all spectra from a particular scan of a scanning-DOAS instrument are contaminated with (volcanic) $SO_2$ absorption structures. Additionally it allows us to investigate under which conditions scans occur with $SO_2$ absorptions present in all viewing directions.

## 2    Background spectra for scanning DOAS instrument networks at volcanoes

Most techniques for the remote-sensing of trace gases in volcanological environments are based on Bouguer-Beer-Lambert's

law. Among these techniques are COSPEC, DOAS and the $SO_2$ camera technique (Mori and Burton, 2006; Bluth et al., 2007). For the simple case of only one absorber Bouguer-Beer-Lambert's law can be written as:

$$I(\lambda) = I_0(\lambda) \cdot e^{-S \cdot \sigma(\lambda)} \tag{1}$$

where $\lambda$ is the wavelength, $S$ is the trace-gas CD, the integral of the absorber's concentration along the light-path and $\sigma(\lambda)$ is the absorber's cross-section. $I_0(\lambda)$ (which in the case of sunlight measurements is called the FRS) is the radiation intensity

before and $I(\lambda)$ is the radiation intensity after traversing the medium of interest. If $\sigma(\lambda)$, $I(\lambda)$ and $I_0(\lambda)$ are known, Equation 1 can be solved for $S$ in the case of one absorber:

$$S = -\frac{1}{\sigma(\lambda)} \cdot \ln \frac{I(\lambda)}{I_0(\lambda)} = \frac{\tau}{\sigma(\lambda)} \tag{2}$$

In the case of several absorbers $i$, the CDs $S_i$ can be retrieved by fitting the absorption cross-sections $\sigma_i(\lambda)$ to a the optical density spectrum $\tau(\lambda)$. For sunlight measurements, the FRS $I_0(\lambda)$ is needed to remove the shape of the incident solar radiation,




in particular the strongly structured Fraunhofer lines, and possible instrumental structures from the measurement. In DOAS, not the optical density $\tau$, but the differential optical density is used to determine $S$. This can, for example, be achieved by including a polynomial in the retrieval or by high-pass filtering of all spectra (Platt and Stutz, 2008).

Note that the evaluation procedure always relies on the ratio of spectra. While the measurement spectrum is given by Equation 1 we can also write down the FRS explicitly:

$$I_0(\lambda) = I_{solar} \cdot e^{-S_0 \cdot \sigma(\lambda)} \tag{3}$$

and after replacing $I_0$ by $I_{solar}$ in Equation 1 we obtain for the ratio:

$$\frac{I(\lambda)}{I_0(\lambda)} = \frac{I_{solar} \cdot e^{-S \cdot \sigma(\lambda)}}{I_{solar} \cdot e^{-S_0 \cdot \sigma(\lambda)}} = e^{-\sigma(\lambda) \cdot (S - S_0)} \tag{4}$$

which leads to:

$$-\frac{1}{\sigma(\lambda)} \cdot \ln \frac{I(\lambda)}{I_0(\lambda)} = S - S_0 \tag{5}$$

The great advantage of the ratio method is that the highly structured solar Fraunhofer spectrum ($I_{solar}(\lambda)$) is eliminated in the evaluation. The potential weakness of the method is that it actually always determines the difference of two CDs, the so-called differential Slant Column Density (dSCD). Note that this property also helps to eliminate potential stratospheric contributions to the trace gas column (if the SZA is sufficiently constant between the measurement of $I(\lambda)$ and $I_0(\lambda)$), in addition possible instrumental structures are also removed. Because of these properties the ratio method is almost universally used in scattered light DOAS applications. However, one has to make sure by proper choice of $I_0(\lambda)$ that $S_0$ is negligible compared to $S$.

At volcanoes, during traverse measurements or campaign based scanning measurements a gas-free FRS $I_0(\lambda)$ is typically obtained by choosing a spectrum recorded with a viewing direction that is believed not to intersect the plume. However, the choice of FRS is more difficult for automatised scanning DOAS networks like NOVAC (Galle et al., 2010) or the FLAME network (Burton et al., 2009), since it is not always clear whether the FRS contains significant volcanic $SO_2$ absorption structures or not.

Galle et al. (2010) suggested for NOVAC to use a zenith-looking spectrum as FRS for the DOAS evaluation. Possible contamination of the FRS is taken into account during the evaluation by subtracting an offset $SO_2$ CD (that corresponds to $-S_0$) from the derived CD $S$. Based on preliminary tests on measurements at a few volcanoes, the authors suggested to use the average of the lowest 20% $SO_2$ CDs from the valid retrievals in each scan as the offset CD $-S_0$ but other options (e.g. using the lowest $SO_2$ CD obtained in a scan) are possible as well. This offset value is determined individually for each scan (i.e. recordings of spectra from one horizon to the other) and subtracted from all $SO_2$ CDs of the respective scan. However, if all spectra of a scan are influenced by $SO_2$ absorption (i.e. $S_0$ is not negligible compared to $S$) subtracting the offset will lead to an underestimation of the $SO_2$ CDs. Therefore, this approach can lead to a systematic underestimation of the $SO_2$ emission rate, if $SO_2$ absorption structures are present at all viewing directions of the instrument.

For the FLAME network, Salerno et al. (2009) investigated the use of a modelled background spectrum for the DOAS evaluation of $SO_2$. The authors noted that wide volcanic plumes, which may cover the entire field of view of the instrument and



prevent acquisition of a plume-free reference spectrum, are relatively frequently observed at Etna, Italy. The authors recorded spectra of calibration cells (with known $SO_2$ content) and tuned the parameters of the DOAS evaluation in order to reproduce the known $SO_2$ CD of the cells. This makes this approach rather labour intensive and it does not appear to be practical for instruments which are already installed at remote locations. Salerno et al. (2009) used three different values for the full width

at half maximum (FWHM) of the instrument line function (ILF) for the convolution of the different trace-gas absorption cross-sections and the convolution of the high-resolution Solar atlas spectrum (i.e. the FWHM for the $O_3$ convolution was different from the FWHM for the $SO_2$ convolution). However, the ILF is an instrument property. While there are influences, e.g. variations of the FWHM over the detector or variations with instrument temperature, it does not depend on the trace gas itself, and therefore, there is no physical reason to encounter three different FWHM values of the ILF.

In our new evaluation scheme, we followed the approach to model the FRS on the basis of a Solar atlas instead of measuring it on site, instrumental properties are retrieved from the measurement data itself. We used the Chance and Kurucz (2010) Solar atlas spectrum as a basis to model the FRS. The modelled FRS $I_{0,model}(\lambda)$ for the DOAS evaluation is obtained by convolving the high resolution Solar atlas $I_K(\lambda)$ with the ILF $H(\lambda)$ as a convolution kernel:

$$I_{0,model}(\lambda) = I_K(\lambda) * H(\lambda) = \int I_K(\lambda - \lambda') \cdot H(\lambda') \, d\lambda' \tag{6}$$

In our approach we used the same ILF for the convolution of the high-resolution Solar atlas spectrum as well as for the convolution of all trace gas cross-sections. Unfortunately there are only records of the ILF at room temperature available for most NOVAC instruments. This introduces an additional error source, since it is known that the ILF varies with instrument temperature (Pinardi et al., 2007). All reference cross-sections were convolved using the 334.15 nm line of a mercury emission lamp which was recorded at room temperature as a convolution kernel.

In reality, the recorded signal due to the incident solar radiation is not only influenced by the spectral resolution of the spectrometer but also by the wavelength dependent efficiency of the detector and the efficiency of the spectrometer's grating or by the instrument's optical system. We combine these wavelength dependent effects in a factor $Q(\lambda)$ (neglecting detector effects like offset and dark current which were corrected beforehand) and describe a measured spectrum as:

$$I_{0,measured}(\lambda) = (I_{incident}(\lambda) * H(\lambda)) \cdot Q(\lambda) \tag{7}$$

Actually only the high frequency variations (in wavelength) of $Q(\lambda)$ need to be corrected for in the retrieval, since slow variations are eliminated by the high-pass filtering inherent to the DOAS technique as explained above. Burton and Sawyer (2016) additionally mentioned small variations between different pixels. In their manuscript, these variations are taken into account by characterizing them with help of a deuterium lamp (i.e. recording a so-called flat spectrum).

Here we use a different approach and include the above mentioned wavelength dependent effects $Q(\lambda)$ and the pixel-to-

pixel variations as a pseudo-absorber in the DOAS retrieval. Assuming that $I_K(\lambda)$ is an ideal representation of the incident radiation $I_{incident}(\lambda)$ (i.e. $I_K(\lambda) = I_{incident}(\lambda)$) we are left with a wavelength-dependent residual structure when calculating the optical density $\tau$ using Equations 6 and 7:

$$\tau = -\log \frac{I_{0,measured}(\lambda)}{I_{0,model}(\lambda)} = -\log Q(\lambda) \tag{8}$$



Two pseudo-absorbers were included in the DOAS fit scenario in order to account for these instrumental effects. Information about these absorbers were obtained from the spectra themselves by using a PCA on the residuals from a DOAS fit. We interpret the first principal component to be caused by detector effects as given by $Q(\lambda)$ in Equation 8. Including the second principal component in the DOAS fit greatly improved the stability of the DOAS fit, in particular for the instruments installed at Tungurahua. This second principal component appears to account largely for variations of the instrument calibration and temperature induced changes of instrument properties. However, the principal components could additionally contain structures from the Chance and Kurucz (2010) Solar atlas as suggested by Burton and Sawyer (2016). This Solar atlas is based on measured spectra synthesized from 2 different measurement platforms and corrected for atmospheric absorption lines.

## 3 Data evaluation

### 3.1 Settings of the DOAS retrieval

All spectra were evaluated for $SO_2$ using the DOASIS software package (Kraus, 2006). First all spectra were corrected for dark current and offset by subtracting a dark spectrum, which is recorded using the same parameters (number of co-added spectra and exposure time) as for the measurement spectra but with the telescope pointing to the ground, where the field of view of the spectrometer is blocked by a closed window of the scanner. Afterwards spectra that were under- or overexposed were removed from the evaluation. This was done in two steps: First we removed all spectra for which the highest exposure was below 12% or above 92% of the maximum number of counts in the complete spectrum (500 and 3800 of 4096 counts, respectively, for a single exposure). These limits refer to spectra corrected for dark current and offset. Limiting the maximum exposure over the complete spectrum (not just the part used for $SO_2$ retrieval) served to prevent problems due to blooming effects. Second, after investigating the fit quality for both retrieval methods (see below) we additionally excluded all spectra with a maximum intensity below 5% or above 85% in the $SO_2$ retrieval wavelength range from further processing. The latter maximum value was chosen since the $\chi^2$ of the retrieval largely increased at higher intensities due to detector non-linearity.

After a wavelength-calibration (by comparing the spectrum with the Fraunhofer lines of the Chance and Kurucz (2010) Solar atlas spectrum) spectra were evaluated for $SO_2$ using a DOAS fit. The DOAS evaluation was performed in the wavelength range between 310 - 326.8 nm. In order to create the principal components (for which we need vectors with the same length) we chose the channels corresponding to these wavelengths once at the beginning, and kept the channels of the fit range fixed throughout the entire evaluation process.

All spectra were evaluated using two different methods:

- Method A: An evaluation similar to the regular NOVAC evaluation. This method used a spectrum that was acquired by the instrument immediately before the scan with a scan angle of $0°$ (smallest deviation from the zenith direction, see Galle et al., 2010, for the exact definition of the scan angle) as FRS. After evaluation of a complete scan through the sky, which means recording spectra from one horizon to the other horizon, an $SO_2$ CD offset was calculated and subtracted



from all $SO_2$ CDs of the respective scan. In this manuscript, the lowest $SO_2$ CD from each scan was used as the offset value.

– Method B: An evaluation using a modelled background spectrum, based on a high-resolution Solar atlas spectrum as an FRS. In this case the same FRS was used for the evaluation of all spectra. It was calculated on the basis of the Chance

and Kurucz (2010) Solar atlas by convolving the high-resolution Solar atlas spectrum with the ILF of the respective spectrometer (see Equation 3, above). First a DOAS evaluation using this fit scenario was performed to create a set of residual spectra which were analysed with help of a principal component analysis. In a second evaluation round the first two principal components were included in the fit scenario as pseudo-absorbers in order to account for instrumental effects (see Equations 4 and 5, above). The results of the second run were used in order to investigate the relative

difference between the two methods.

As mentioned above, the 334.15 nm peak of a mercury line spectrum recorded at room temperature was used as a representation of the ILF. Based on the respective FRS, a Ring spectrum was calculated and included in the fit in order to account for the Ring effect, the filling of the Fraunhofer lines of the solar spectrum (Shefov, 1959; Grainger and Ring, 1962). The Ring spectrum is calculated using a method that is implemented in DOASIS which is based on Bussemer (1993).

The DOAS fit approach was (except for the FRS and the additional pseudo-absorbers) - identical for methods A and B: To account for trace gas absorption in the atmosphere one $SO_2$ cross section (recorded at 298K by Vandaele et al., 2009) and two $O_3$ cross sections (recorded at 221K and 273K by Burrows et al., 1999) were included in the DOAS fit. The $O_3$ cross section with a temperature of 273 K was orthogonalised with respect to the lower temperature $O_3$ cross section in the DOASIS software. To account for small inaccuracies in the wavelength calibration, the FRS and the Ring spectrum as one set and all

trace gas reference cross-sections as another set were allowed a wavelength shift and squeeze with respect to the measurement spectrum for Method A. A shift of $\pm0.2$ nm was allowed and the spectrum was allowed to be stretched/squeezed by +-2% in order to improve the fit. Since the modelled FRS spectrum (Method B) is synthetic, the calibration is inherited from the Solar atlas. Given the accuracy of the calibration of the Solar atlas $\leq 3.2 \cdot 10^{-4}$ nm (Chance and Kurucz, 2010), we assume that it is correct for our purposes. Therefore the FRS, the Ring spectrum and all trace gas absorption cross-sections were only allowed

to be shifted and squeezed as one set in Method B. Some of the instruments had a "hot detector pixel", a pixel which always showed a much higher signal. For Method A this was not a problem, since a similar signal is typically found in the FRS. To exclude these hot-pixels from the fit in Method B an additional absorber that is zero everywhere except at the location of the hot-pixel (where its value was chosen as 1) was included in the fit. A DOAS polynomial of $3^{rd}$ order was used to remove broadband variations in the spectrum. In order to take a possibly remaining offset into account (after the dark spectrum removal, for

example due to instrumental stray-light), an additional constant in intensity space was allowed in the retrieval.

After an initial DOAS fit the retrieved trace gas CDs were used as input parameters for a saturation and $I_0$ correction of the $SO_2$ and the $O_3$ absorption cross-sections. Both the $I_0$-effect (for highly structured light sources) and the saturation effect (for absorbers with high optical densities) are due to narrow structures that cannot be resolved by the spectrometer and the fact that the exponential function in the Bouguer-Beer-Lambert's and the convolution do not commute (Wenig et al., 2005).



The correction of both effects are standard procedures in DOAS evaluations (Platt et al., 1997; Platt and Stutz, 2008). Both $O_3$ cross-sections were corrected for the saturation effect using the CD of the (non-orthogonalized) $O_3$ cross-section recorded at 221 K.

### 3.2 Calculation of the $SO_2$ offset value for Method A

As discussed above, it cannot be ruled out that the spectrum used as FRS in Method A is contaminated with $SO_2$. One approach to (partially, see above) correct for this contamination is to calculate an $SO_2$ offset value. This value is calculated for each scan and subtracted from all $SO_2$ CDs of the respective scan. Deviating from the approach of Galle et al. (2010), we used the lowest $SO_2$ CD of each scan as the $SO_2$ CD offset instead of using the average over the $SO_2$ CD of several spectra. As the offset value is based on only one single spectrum it is important to remove spectra where the $SO_2$ fit failed completely from the results

before calculating the offset value.

    We therefore only used viewing directions that were not influenced by obstacles in the field of view and where the intensity was adequate, for further evaluation. This meant, for example, that for instrument D2J2201 at Nevado del Ruiz only spectra with a scanning angle between -72° and 86° were allowed (the scan angle is defined from -90° to +90° clockwise for an observer looking from the instrument towards the volcano). If no viewing directions influenced by obstacles were identified, we only

excluded the two lowest viewing directions for these instruments (e.g. scan angle of ±90°). Limiting the viewing directions for the calculation of the offset is a trade-off. Viewing directions with obstacles (for example mountains or buildings) in the field of view obviously lead to erroneous DOAS fit results. However, excluding too many viewing directions could influence the results, since we might in some cases exclude viewing directions which are interference free.

    We also removed spectra from further evaluation, where the DOAS fit from Method A clearly failed and had a $\chi^2$ above

0.05. $\chi^2$ was calculated for all pixels of the evaluation range and typically had values between $1 \cdot 10^{-3}$ and 0.01. Therefore, this threshold only removed a small fraction below 1% of the remaining spectra from further evaluation. Afterwards the lowest $SO_2$ CD of each scan was chosen as the offset value and subtracted from all $SO_2$ CDs of the respective scan.

### 3.3 Principal Component Analysis for Method B

The Principal Component Analysis (PCA, Pearson, 1901; Smith, 2002) is a statistical technique that can be used to transform

a set of vectors (in our case the remaining residual structure of the DOAS fit) into a set of orthogonal vectors and also provides immediately time-series for the magnitude of each of the vectors. These orthogonal vectors are chosen in a way, that a sequence of $n$ principal components provides the best possible linear approximation of the residual data using an euclidean norm (Hastie et al., 2001). The PCA technique was first applied in DOAS applications by Ferlemann (1998). Li et al. (2013) retrieved $SO_2$ from OMI satellite data with help of PCA. The PCA was used by Lampel (2014) to identify problems in the spectral

evaluation of Multi-Axis DOAS and cavity-enhanced DOAS measurements. We performed a PCA on the residuals of the initial DOAS fit with the modelled FRS in order to take instrumental effects into account (see Equation 8). Different from usual PCA applications we did not remove the mean value from the spectra, since we are interested in all systematic variations of the residual spectra from the zero value (which would be the ideal case).



In order to find mainly instrumental effects and exclude other problems in the residuals (e.g. large $O_3$ CDs, objects in the light-path), even more restrictive criteria had to be fulfilled by the spectra included in the PCA:

- Only spectra with a scan angle in the range of -75° to +75° were included in order to avoid influences by very long atmospheric light paths and errors due to topographic features or buildings/vegetation in the light-path.

- Only spectra recorded at SZAs below 60° were used, to exclude spectra with large stratospheric $O_3$ CDs.

- Only spectra with a $\chi^2$ below 0.01 for Method A (using a spectrum recorded with the same instrument before the scan as FRS) were included in order to exclude spectra that are already problematic in a regular DOAS retrieval.

- A more restrictive selection criterion for the intensity of the spectrum was chosen. Only spectra with a maximum number of counts between 32% and 78% (1333 or 3200 counts for a single exposure after dark current correction) of the
maximum possibly number of counts over the entire spectrum were allowed for the PCA.

- Only spectra that were not influenced by $SO_2$ (i.e. $SO_2$ CD below two times the DOAS retrieval error) were allowed for the PCA. This was assured by a DOAS fit using a modelled spectrum as FRS without including additional PCA pseudo-absorbers (see Figure 1).

There are some pitfalls which can make the application of the PCA non-trivial. The spectra that are analysed in the PCA
should not include any real $SO_2$ absorption features, because, if they did this would introduce a potential negative $SO_2$ offset. These features would show up in the residual structure and thus influence the principal components and ultimately lead to unreliable fit results. Li et al. (2013) assured this criterion by selecting an $SO_2$ free reference sector. For our data set we have to use a different approach to only select spectra without $SO_2$ structures for the PCA. We chose spectra from times with only little degassing activity to create the PCA. Since this was difficult (in particular at Nevado del Ruiz) rather than relying on
guesses about the $SO_2$ CD we used an additional $SO_2$ fit with a modelled FRS to select gas free spectra. Only spectra where the absolute value of the $SO_2$ CD was smaller than twice the DOAS fit error were considered in the PCA. Using a similar argument, including an $SO_2$ absorption cross-section in the DOAS fit used for the PCA can lead to problems. Inaccuracies in the DOAS fit with a modelled FRS (due to the same detector structures that we want to find with help of the PCA) might lead to a false $SO_2$ signal (with positive or negative sign). The fit might find these structures and thus remove them from the residual
spectrum. Thus they are missing in the principal component which is later included in the DOAS fit. In this case the principal component would not only describe the instrumental effects, but it would add/subtract $SO_2$ features from the spectrum and thus lead to an additional error of the $SO_2$ CD. It is therefore crucial not to include $SO_2$ in the DOAS fit that is used to find the principal components for the next iteration. Therefore, we made two DOAS fits with the modelled FRS. The first fit had an $SO_2$ absorption cross-sections included and was used to select spectra suited for the PCA. The second fit did not include an
$SO_2$ absorption cross-sections and was used to create the residuals.

At Nevado del Ruiz we selected 7 days in September/October of 2010 as our sample data set for the PCA, the increasing activity after this time made it difficult to find gas free days. At Tungurahua the situation was quite different, the volcano had





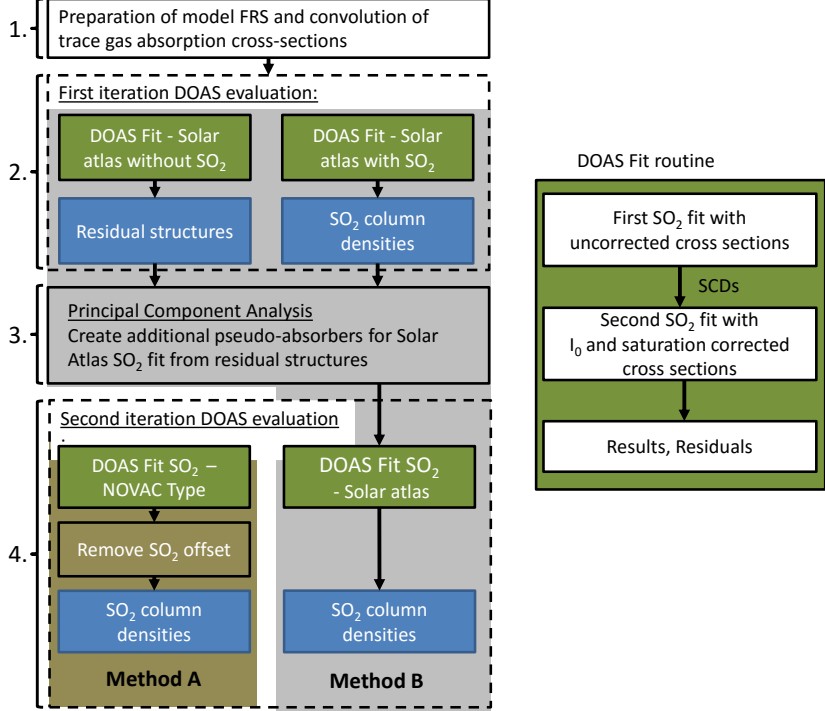

**Figure 1.** Flow chart of the evaluation steps that were used in order to create the $SO_2$ column densities from Methods A and B.

periods of higher activity alternating with times with very low or no degassing at all. The instruments at Tungurahua also showed a drift of instrument calibration and we thus performed the PCA for each year individually. For each instrument and each year at Tungurahua we chose 10 days as our sample data for the PCA. These 10 days were distributed over the year (at times of low volcanic degassing activity) in order to find long term variations of the principal components. For example in 5   2009 we chose five days in January and five days in October for the PCA. For each PCA (that means one for each instrument at Nevado del Ruiz and one per instrument and per year at Tungurahua) typically more than 10.000 residual spectra were evaluated.

### 3.4   Implementation of the new algorithm

In summary the evaluation of data in this study encompassed the following steps (which are also shown in Figure 1):

10   1. Preparation of the modelled FRS and the gas absorption cross-sections:

   (a) A gas-free spectrum, recorded with a small SZA, was wavelength-calibrated by comparing it with the Fraunhofer lines of the Chance and Kurucz (2010) Solar atlas spectrum. This spectrum was subsequently used as the wavelength grid for this instrument, i.e. all trace gas cross-sections and the Ring spectrum were sampled at the wavelength points where this spectrum was sampled.





(b) The Chance and Kurucz (2010) spectrum was convolved using the ILF of the instrument and interpolated to match the wavelength grid from step 1a, this is our modelled FRS for Method B. The Ring spectrum for Method B was calculated from the modelled FRS.

(c) Two $O_3$ and the $SO_2$ absorption cross-sections were convolved using the ILF of the instrument and the same wavelength grid as above. In order to speed up the evaluation all cross-sections were pre-convolved with saturation and $I_0$ correction using different input SCDs and saved.

2. Spectra were evaluated using two fit scenarios which both use the modelled FRS. The first fit scenario includes the $SO_2$ cross-section, this fit scenario is used to select spectra with negligible $SO_2$ content for the PCA in Step 3. The second fit scenario does not contain $SO_2$, it is use to create the residual structures, which are later used in the PCA. After an initial round – to determine estimates of the $O_3$ and $SO_2$ CDs for the $I_0$ and saturation correction – the $I_0$ and saturation corrected absorption cross-sections (from step 1c) were loaded and a second DOAS fit was performed using $I_0$ and saturation corrected cross-sections (the $SO_2$ CDs and residual structures from these corrected fits were used in Step 3).

3. The residual structures of the DOAS fit (without $SO_2$ in the fit scenario) were examined using the PCA as described in subchapter 3.3. The spectra which were analysed with the PCA were selected with help of the DOAS fit that included $SO_2$ (from Step 2).

4. All spectra were evaluated using the DOAS method as described in Step 2 for a second time. This time two different FRS were used : Method A: A spectrum that was measured with the same instrument directly before the scan (recorded with minimal scan angle) was used as FRS, a Ring spectrum for Method A was calculated from this FRS. All trace gas cross-sections (two $O_3$ as well as one $SO_2$) were included in the fit. For Method A an offset value was calculated for each scan and subtracted from the $SO_2$ CDs of each viewing direction (see 3.2 for details). Method B used the modelled FRS. All trace gas cross-sections ($O_3$ and $SO_2$) and the first two principal components from Step 3 were included in the fit scenario. For instruments with a hot-pixel this was included for Method B as well. $I_0$ and saturation corrected cross-sections were used for Method A and Method B .

# 4 Results

At Nevado del Ruiz we evaluated spectra recorded between 1 January 2010 and 30 June 2012 from two NOVAC stations: Alfombrales (Instrument: D2J2201) and Bruma (D2J2200) (see Lübcke et al. (2013) for a map of the stations and Table 1 for their location with respect to the volcano). After evaluating the spectra, we found that the instrument's GPS antennas occasionally reported erroneous times which lead to offsets in the time stamps of spectra of up to 55 minutes. Before selecting spectra for the final results (Section 4.3 - 4.6), we corrected for possible time offsets with help of a Langley plot of the $O_3$ CDs (see Appendix A for details).





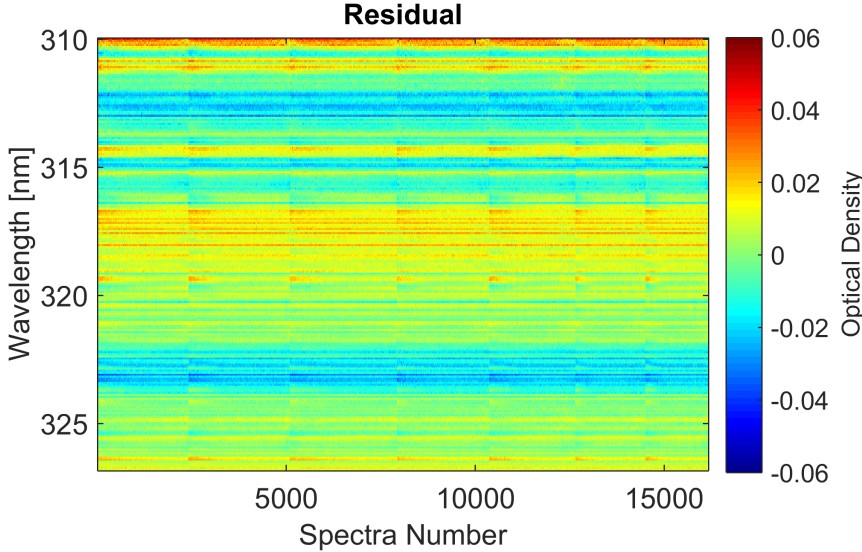

**Figure 2.** Time-series of the residuals optical density (colour-coded) of a DOAS retrieval using a modelled background spectrum. The figure shows data from the 7 days (5-11 September 2010) used in the PCA (without $SO_2$ included in the fit) for instrument D2J2201 at Nevado del Ruiz. The residuals are shown in chronological order. The discontinuities in horizontal direction are at the transition from one day to another. It can be easily seen that the residual structures are largely identical in all evaluations.

At Tungurahua spectra covering three years between January 2009 and December 2011 were evaluated for three stations: Pillate (D2J2140), Bayushig (I2J8546) and Huayrapata (I2J8548). A map of the different NOVAC stations at Tungurahua can be found in Galle et al. (2010).

### 4.1 Structure and variation of the principal components

5 A time-series of the residual structures for the spectra used in the PCA for instrument D2J2201 is shown in Figure 2. It can be clearly seen that there is a dominating constant structure apparent in all spectra. This residual structure is quite similar for all instruments, as can be seen from Figure 3, which shows the first two principal components of the residual structures for all instruments included in this study. While the principal components are not exactly the same, similar broadband as well as narrowband features can be observed for all instruments, in particular for the first principal component.

10 For a better comparison with other absorbers the apparent optical density of the respective principal component (i.e. the fit coefficient multiplied with the peak-to-peak optical density of the pseudo-absorber) is shown in Figures 4 and 5.

At Nevado del Ruiz, we observed that the first principal component is mostly constant with only little temperature variations while the second principal component usually shows stronger temperature dependent variations. This can be seen by looking at the peak to peak optical density of the first and second principal component, shown in Figure 4 for instrument D2J2201.



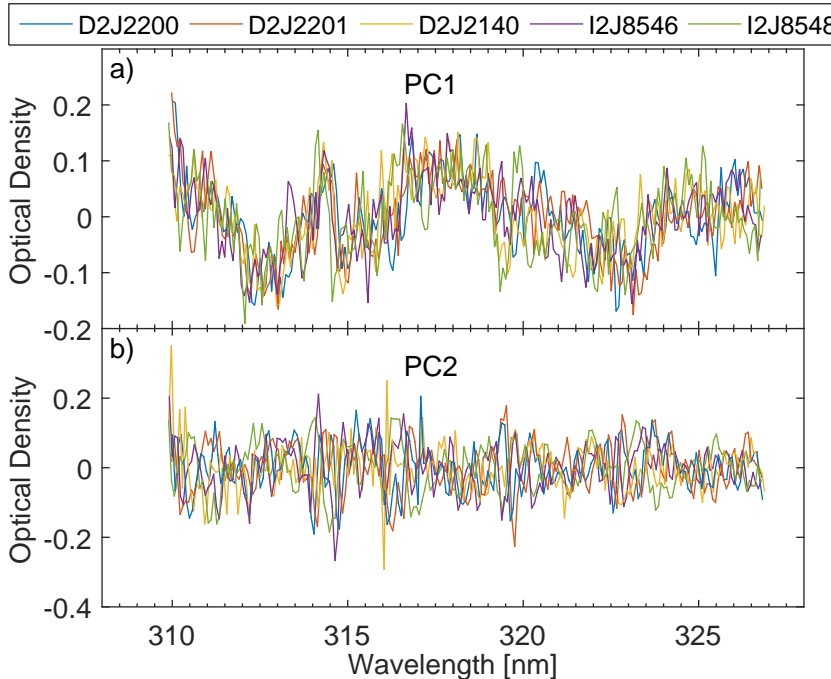

**Figure 3.** The first two principal components that were included in the Solar atlas evaluation shown for all five instruments at Nevado del Ruiz and Tungurahua. For Tungurahua the principal components from 2010 are shown.

The behaviour of the instruments at Tungurahua is more complex, see Figure 5. Both principal components (for all three instruments at this volcano) show a time dependency, a temperature dependency of the principal components can only be observed for some of the instruments.

We interpret the first principal component as the factor $-\log(Q(\lambda))$ from Equation 8 while the second principal component 5 appears to take temperature induced or time dependent variations of the instrument into account.

### 4.2 DOAS fit example

An example of an $SO_2$ fit using a modelled background spectrum is shown in Figure 6. The spectral signature of $SO_2$ ($SO_2$ CD of $1.63 \times 10^{17}$ [molecules/cm$^2$]) can be easily identified in the fit. The residual is unstructured and has a peak to peak value of roughly $1.5 \times 10^{-2}$, this value is comparable to a regular DOAS $SO_2$ fit using an FRS measured with the same instrument 10 from the same scan. Note that the first principal component was found in the fit extraordinarily well while the second principal component does not contribute to the spectrum significantly in this case, which is a spectrum recorded at $16.3°C$.



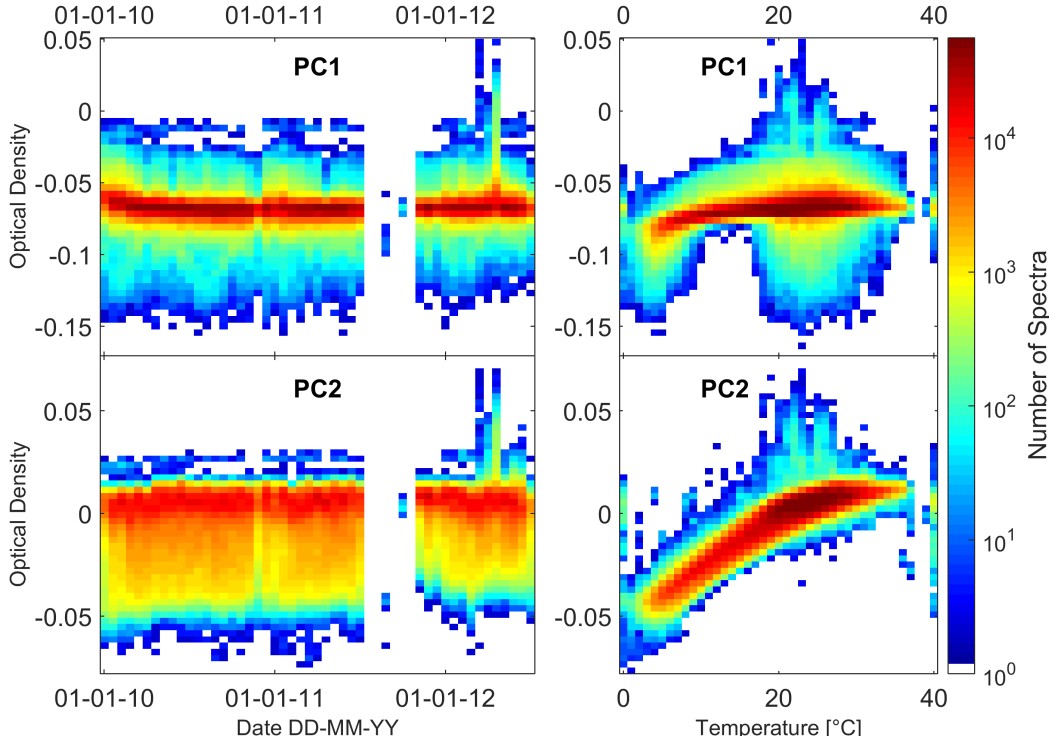

**Figure 4.** Peak-to-peak apparent optical density (e.g. peak-to-peak optical depth of the principal component times the fit coefficient) of the first and second principal components for instrument D2J2201 at Nevado del Ruiz. In the left column the principal components are shown as a function of time, on the right side as a function of spectrometer temperature. It appears that PC1 describes a constant apparent spectral feature of the instrument while PC2 describes a temperature dependent effect.

### 4.3 Comparison of the SO$_2$ column densities from Method A and B

Before directly comparing the SO$_2$ CDs of the two Methods, we first investigate how well Method B performs if there is no absorbing gas in the light-path (i.e. if the retrieval yields zero when no gas is present). We assessed this by manually choosing gas free days and looking at the distribution of the SO$_2$ SCDs. Besides low SO$_2$ CDs from both Method A and B, another

5  criterion to identify gas free days are the variations of the SO$_2$ CDs within one scan, which are typically less structured if no gas is present. At Nevado del Ruiz fewer days (73 or 137 for D2J2200 or D2J2201, respectively) were available in the entire data set due to strong activity. At Tungurahua more data was available, since periods with volcanic activity or no degassing both occur frequently. Examples of the diurnal variation of the SO$_2$ CDs for both instruments at Nevado del Ruiz during gas free days are shown in Figure 7. In contrast to the other instruments (at both volcanoes) instrument D2J2200 (Figure. 7)

10  showed a clear variation of the SO$_2$ CD (as derived by Method B) over the course of a day, with enhanced CDs during the evening. However, the histogram on the right-hand side (which was created for all gas free days) shows that even though the



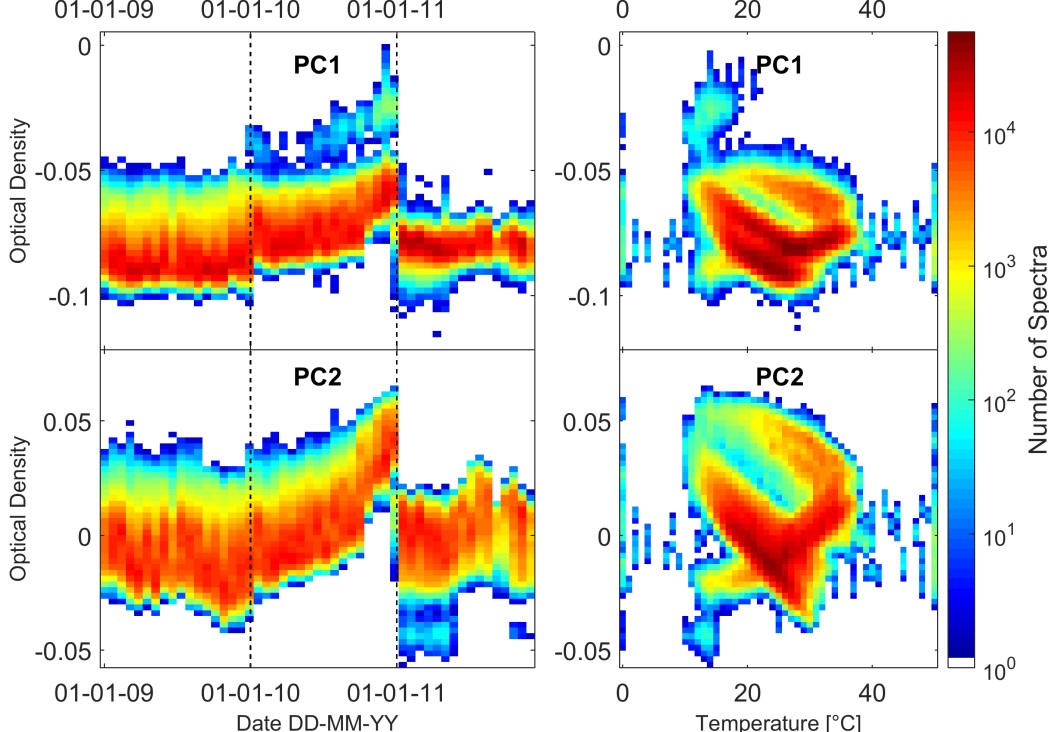

**Figure 5.** Peak-to-peak apparent optical density of the first and second principal components, as Figure 4, but for instrument I2J8548 at Tungurahua. The vertical dashed lines indicate the start of a new time intervals for which a new set of principal components was calculated. In contrast to the case shown in Figure 4 here both, PC1 and PC2 show a temporal drift but a rather chaotic temperature dependence. It is interesting to note, that the instruments at Tungurahua show behaviour similar to this figure. While the spectrometers at Nevado del Ruiz both show a behaviour similar to the one shown in Figure 4

increase towards the evening is clearly visible it only influences a very small fraction of the data set. The other instruments (an example is D2J2201 in Figure 7, bottom) only show deviation from the zero value during the first or last scan of each day. At these times Method A shows larger variation within the scan as well. The mean $SO_2$ CD for gas free periods are between $-7 \times 10^{15}$ [molecules/cm$^2$] and $1.5 \times 10^{15}$ [molecules/cm$^2$], with a standard deviation of $2.7 - 3.6 \times 10^{16}$ [molecules/cm$^2$]. The values of the mean $SO_2$ CD and the standard deviation for all instruments are given in Table 1 in the Appendix.

Besides the zero value, it is also interesting to compare the dSCDs of the two methods. That means, removing an offset $SO_2$ CD from each scan for both methods (we only do this for Method B for this specific comparison). The $SO_2$ offset for Method B was created in the same manner as for Method A.

As discussed in detail below, the $SO_2$ CDs derived by Method B are frequently considerably larger than those derived by Method A. In order to study the ability of both methods to derive the variation of the $SO_2$ CD within one scan we subtracted an





**Figure 6.** Example SO$_2$ DOAS fit of instrument D2J2201. The measurement spectrum was recorded on 5 March 2010 at 16:01 [UTC]. Note that the model functions are shown in red while the model + the residual (i.e. the measurement) are shown in blue.



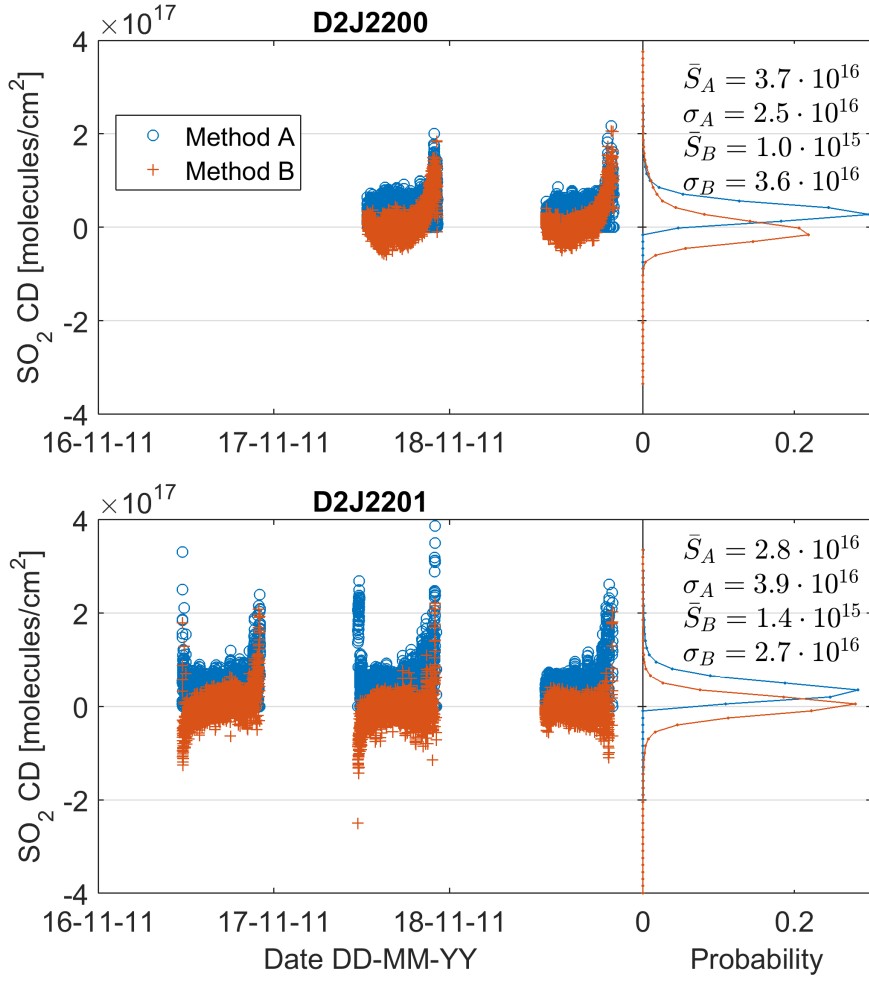

**Figure 7.** $SO_2$ column densities for days with presumably no $SO_2$ apparent in the light path for the two instruments at Nevado del Ruiz. The histogram on the right side shows the distribution of the values of Method B for all gas free days in the data set. An increase in $SO_2$ CDs in the early morning and towards the evening can be observed. However, the histogram shows, that this only influences a small fraction of the data.

offset from the data derived by Method B as well and plotted the result in Figure 8, which shows a two-dimensional histogram of the $SO_2$ CDs from Method A (in y-direction) and Method B (in x-direction, after offset removal) for the complete data set (January 2010 - June 2012) for instrument D2J2201 at Nevado del Ruiz. We used a bin size of $5 \cdot 10^{16}$ [molecules/cm$^2$] (in x-as well as in y-direction), the colour denotes how often a certain $SO_2$ CD pair exists. The slope of this curve can be interpreted

5 as the relationship between the dSCDs from the two methods. When an offset is removed for both methods, they show a linear relationship. The slope of this curve is not exactly unity, it varies between 0.88 and 1.14 for the different instruments (the values are given in Table 1 in the Appendix). It is not entirely clear, what causes the difference between the two methods. One possible





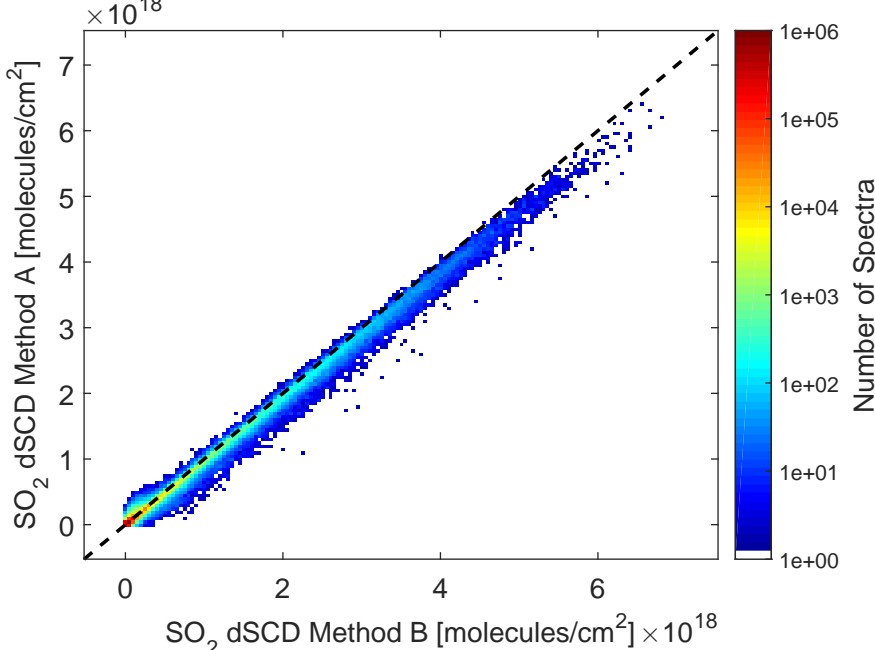

**Figure 8.** 2D Histogram of the $SO_2$ SCDs from the Solar atlas method vs. the SCDs from the NOVAC evaluation for instrument D2J2201 at Nevado del Ruiz. For this figure an offset value was subtracted from the Solar atlas $SO_2$ CDs as well. The dSCDs of both methods show a good linear relationship with a slope close to 1 (0.956 in this case). The black-dashed line shows the identity line.

explanation is that spectrometer stray-light (that should be corrected for by an additional offset in intensity space during the fit) is not corrected for in the same way for Method A and B.

After making sure that Method B performs well for gas free days and that both methods show similar $SO_2$ dSCDs (within a certain error), we trust that Method B works and compare the absolute $SO_2$ CDs from Method A with Method B. One rather
5   extreme example of the difference between the $SO_2$ CDs derived for both methods is shown in Figure 9. This figure shows data from instrument D2J2201 at Nevado del Ruiz recorded on March 6 2012. It can be clearly observed that Method B retrieved much larger $SO_2$ CDs, especially during the middle of the day. At this time the modelled FRS leads to $SO_2$ CDs of up to $5 \cdot 10^{18}$[molecules/cm$^2$] while Method A only shows CDs around $1 \cdot 10^{18}$[molecules/cm$^2$]. It should also be noted that the variations within each scan (which can be identified by the small gaps between data points) show a similar variations for both
10  methods. However, for Method A each scan has one viewing direction with an $SO_2$ CD of 0, since we subtracted an offset value for each scan (see 3.2). Also note that for calculation of $SO_2$ emission rates additional criteria exist, that would have let to discarding most measurements on the day presented in Figure 9.

We also observed days, where both methods agreed nicely. As one example January 11 2012 is shown in Figure 10. Only a part of the day is shown for a better visibility of the variability of the $SO_2$ within each scan.





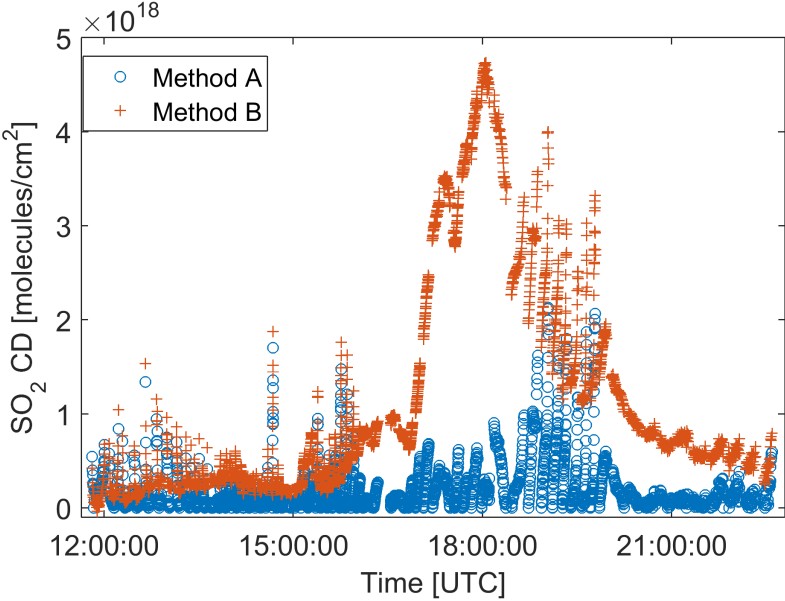

**Figure 9.** $SO_2$ CDs at Nevado del Ruiz on 6.3.2012 from instrument D2J2201. Method A (which is similar to the standard NOVAC approach) leads to much lower $SO_2$ CD when compared Method B (modelled FRS). It should be noted that data from this day would have been rejected by the standard NOVAC evaluation due to very low plume completeness values (see Galle et al., 2010, for details on plume completeness).

A more systematic way to compare the $SO_2$ CDs of many measurements is shown in the histogram in Figure 11. This figure again shows 2D-histograms of the $SO_2$ CDs from Method A (in y-direction) and Method B (in x-direction, this time without removing an offset) for the complete data set (January 2010 - June 2012) for instrument D2J2201 at Nevado del Ruiz. The same bin size as above was used. As can be seen in Figure 11, Method B leads to larger $SO_2$ column densities on a significant

number of spectra. When evaluated by Method A only a negligible number of spectra has larger $SO_2$ column densities than given by Method B. The larger $SO_2$ CDs of Method B are most likely caused by contamination of all viewing directions with $SO_2$ absorption structures (the likely reasons for the contamination will be discussed in Section 5).

### 4.4    $SO_2$ fit error for Method B

The time-series of the $SO_2$ retrieval error (for instrument D2J2201) depicted in Figure 12 shows a more or less constant

distribution of the fit error between January 2010 and February 2012, the majority of spectra have an fit error below $2 \times 10^{16}$ [molecules/cm$^2$]. Much larger $SO_2$ fit errors after March 2012 are caused by strong volcanic activity and large $SO_2$ CDs that lead to non-linearities in the DOAS retrieval (Method A shows increased retrieval errors in this period as well).

The $SO_2$ DOAS fit error of Method B is shown as a function of instrument temperature in Figure 13 and as a function of SZA in Figure 14. In Figure 13 we can clearly observe an increase of the fit error when the temperature falls below 10°C. This

can be explained with the variation of the ILF with instrument temperature. The ILF used for the convolution of the Chance and





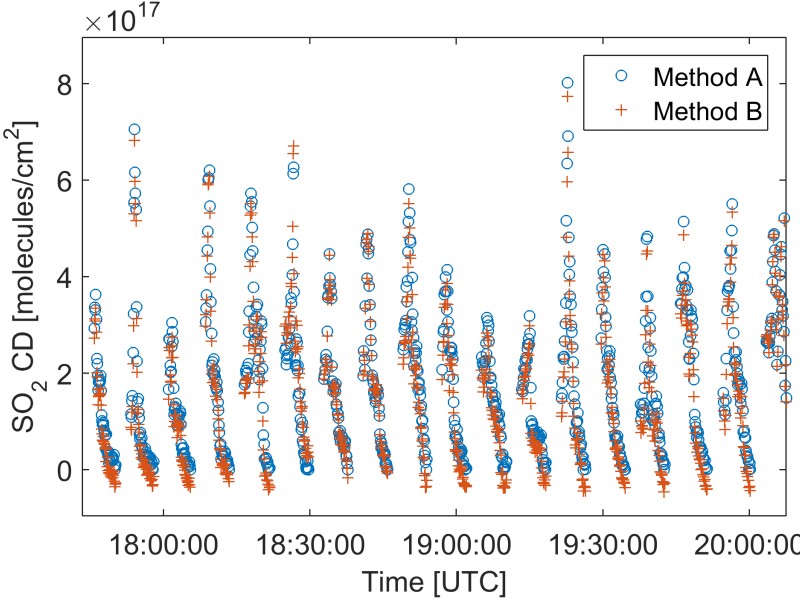

**Figure 10.** Zoom into the variation of the SO$_2$ CD at Nevado del Ruiz from instrument D2J2201 on 11.1.2012. In this example both methods agreee well.

Kurucz (2010) Solar atlas spectrum and the absorption cross-sections was recorded at room temperature. Pinardi et al. (2007) investigated the change of ILF with temperature and reported variations of the ILF of up to 0.1 nm, in particular at temperatures below room temperature.

In Figure 14 we can observe that the SO$_2$ retrieval error largely increases at solar zenith angles above 75°. This behaviour

5  can be caused by a couple of reasons: for one less radiation is available at low SZAs, in particular in the low UV used for the SO$_2$ evaluation, leading to a poor signal-to-noise ratio. Additionally, low SZAs coincide with lower temperatures, in particular in the morning hours. A third reason can be interferences between O$_3$ and SO$_2$ in the DOAS retrieval. Both trace gases have quite similar absorption structures in the UV, large O$_3$ CDs at large SZAs can lead to non-linearities in the absorption or in the photon light-path and thus result in an increased residual structure.




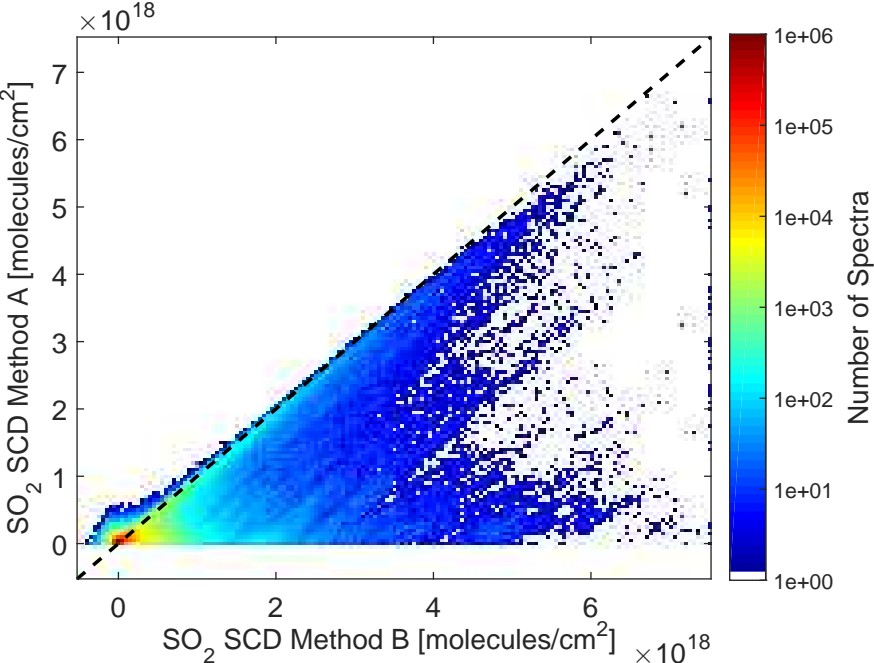

**Figure 11.** 2D Histogram of the $SO_2$ SCDs from Method B vs. the SCDs from Method A for instrument D2J2201 at Nevado del Ruiz. Method B, which uses a modelled background spectrum leads to larger $SO_2$ column densities. The black-dashed line shows the identity line.

## 4.5 Relative difference of the $SO_2$ content from Method A and Method B

Section 4.3 showed that the more commonly applied Method A (removing an $SO_2$ offset in order to correct for $SO_2$ contaminated background spectra) sometimes leads to a difference of the $SO_2$ CD compared to Method B. In order to identify how frequent a significant difference of both methods can be observed, we look at the relative ratio $R$ of the $SO_2$ CDs determined with the two methods:

$$R = \frac{\overline{S_{SO_2}(B)} - \overline{S_{SO_2}(A)}}{\overline{S_{SO_2}(B)}} \tag{9}$$

where $\overline{S_{SO_2}(B)}$ is the average $SO_2$ CD for one scan from Method B and $\overline{S_{SO_2}(A)}$ is the average CD from Method A. Since the slope of the linear fit against the dSCDs of Method B and Method A was not exactly unity (see Section 4.3 and Figure. 8), we multiplied the SCDs from Method B with a correction factor (see Table 1). We only averaged over spectra in a scan that exceed a certain $SO_2$ threshold (for Method B) in order to reduce influences of possible retrieval inaccuracies and to avoid dividing by zero when calculating the relative difference $R$. Additionally, in order to reduce possible errors due to strong ozone absorption, we only used the following spectra for further investigation:



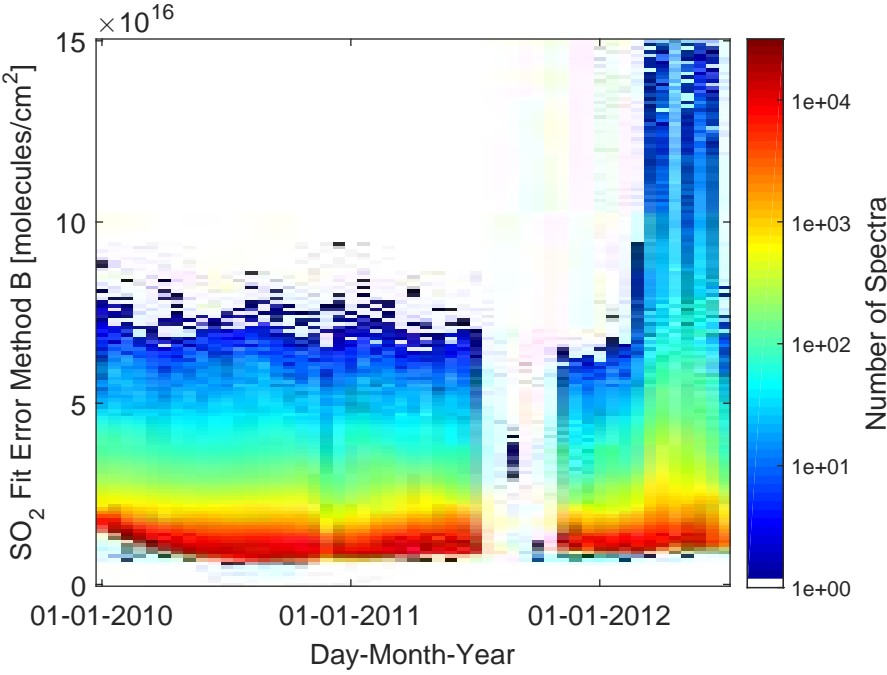

**Figure 12.** 2D-Histogram of the $SO_2$ DOAS fit error from the Solar atlas evaluation as a function time for instrument D2J2201.

– Only spectra with an $SO_2$ CD above $5 \times 10^{17}$ [molecules/cm$^2$] were taken into account for the averaging process. This ensures a robust relative ratio by preventing divisions with values close to zero in Equation 9 and reducing the influence of inaccuracies in the retrieval for low $SO_2$ contents.

– spectra with an Solar Zenith Angle below $70°$, in order to circumvent potential problems due to strong ozone absorption
at lower SZAs.

Figure 15 shows histograms of the relative difference $R$ for Nevado del Ruiz. For Tungurahua the results are shown in Figure 16. The histogram plots show the relative difference for all data (top left) and for different wind speed intervals in the other plots. Wind speeds were taken from the ECMWF (European Centre for Medium-Range Weather Forecasts) database. Wind data is obtained from analysed data products from ECMWF at a spatial resolution of 0.75 deg and a time resolution of
6 h. Data is interpolated to the location of the crater and time of measurement (in this case the original time stamp from the instruments was used).

It can be seen in the top left of Figure 15 that the distribution of the relative ratio has a peak at 0 (e.g. both methods give the same $SO_2$ CD) and a tail that goes up to a relative ratio of 100%.

The other histograms in Figure 15 show the same as Figure 15 (a), however, each histogram only shows data for a specific
wind speed interval. For wind speeds above 10 m/s (Figure 15 (b)) a dominant peak at a relative ratio of 0% can be observed with a few values at a higher relative ratio. This means that both methods essentially give the same result. For wind speeds between



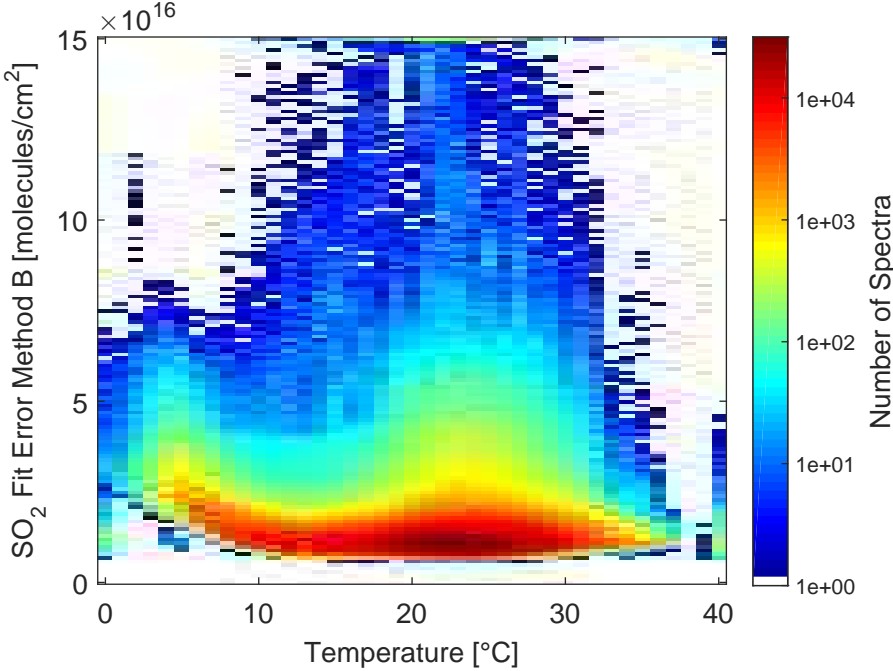

**Figure 13.** 2D-Histogram of the $SO_2$ DOAS fit error from the Solar atlas evaluation as a function of instrument temperature for instrument D2J2201.

5 m/s and 10 m/s we observe a slight increase of larger relative ratio values (c). For wind speeds below 5 m/s (Figure 15 (d)) we see a homogeneous distribution with almost constant values between 0% and 100%. This means that we can observe wide spread plumes which cover the complete FOV of the scanningDOAS instrument more frequently at low wind speeds.

When comparing the histograms for Nevado del Ruiz (Figure 15) and Tungurahua (Figure 16) we can observe that the

5 relative difference is larger at Nevado del Ruiz. At Nevado del Ruiz 21.4% of the scans containing significant $SO_2$ have a relative difference above 0.5, compared to only 7% at Tungurahua. This means that we have contaminated reference spectra more frequently at Nevado del Ruiz. While the relative difference of 0.5 is an arbitrary value, it means only 50% of absolute $SO_2$ CD is detected. One possible explanation for the larger number of contaminated spectra at Nevado del Ruiz is the distance of the instruments from the crater (see Table 1). The instruments at Nevado del Ruiz are installed $\sim 3 - 4$ km from the crater,

10 while the instruments at Tungurahua are installed at more than 8 km distance.

## 5  Conclusions

We developed a new evaluation scheme for volcanic $SO_2$ not relying on any locally recorded Fraunhofer reference spectrum, but on a FRS modelled based on a high-resolution Solar atlas, which makes our method immune against possibly contaminated reference spectra (i.e. FRS containing absorption structures due to $SO_2$). Using this new method we investigated how frequently





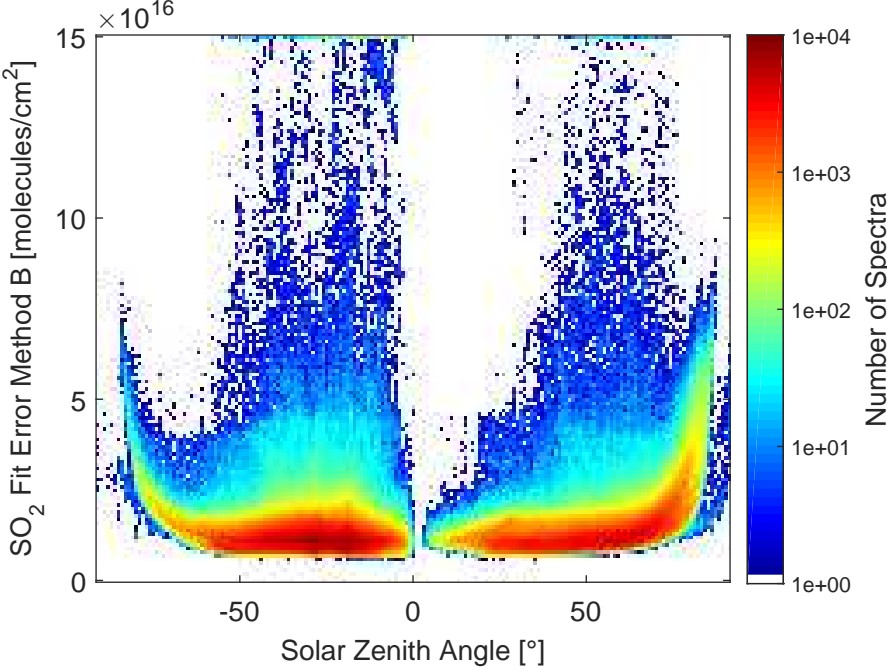

**Figure 14.** Histogram of the $SO_2$ DOAS fit error from the Solar atlas evaluation as a function of solar zenith angle for instrument D2J2201.

contaminated FRS occur for scanning DOAS instruments from NOVAC at the volcanoes Nevado del Ruiz and Tungurahua. We observed that the DOAS retrieval, which used the convolved Chance and Kurucz (2010) Solar atlas spectrum as FRS, typically showed a similar residual structure for all spectra (before including principal components in the fit). A PCA on the residual structures revealed that the first principal component accounts for more than 88% of the variation of the residual structures (for

all instruments), while the combination of the first two principal components typically accounts for even more than 90% of the variation. Each of the further principal components only has an individual contribution below 1% to the residual structure. We interpret the first two principal components as instrumental effects. The first principal component describes the quantum and grating efficiencies of the spectrometer, while the second principal component takes temperature induced variations into account. However, we cannot completely rule out, that these principal components also include structures that are inherent in

the Chance and Kurucz (2010) Solar atlas as suggested by Burton and Sawyer (2016). After including the first two principal components as pseudo-absorbers in the DOAS fit we obtained a fit quality comparable to a regular DOAS $SO_2$ evaluation with an $SO_2$ DOAS retrieval error as good as $1 \times 10^{16}$ [molecules/cm$^2$]. The $SO_2$ fit error shows increased values with low instrument temperatures and high SZA. The temperature dependency of the fit error can be explained by temperature induced variations of the ILF. Taking these effects into account could further improve the evaluation in the future.

We found that the $SO_2$ evaluation based on a modelled FRS (Method B) finds the zero level well and that the dSCDs of this method typically lies within 15% of the dSCDs of a standard DOAS retrieval using an FRS recorded with the same instrument (Method A). Furthermore, the new method allows us to retrieve absolute $SO_2$ CDs. The comparison of the $SO_2$ CDs of the





two methods showed, that Method A in some cases leads to smaller $SO_2$ CD values than Method B. We found that at Nevado del Ruiz 21.4% of the scans containing a significant amount of $SO_2$ in all viewing directions (according to Method B) show much lower $SO_2$ CDs for Method A (factor of 2, which corresponds to a relative ratio of 0.5). At Tungurahua only 7% of the scans have a relative ratio above 0.5. The relative ratio between the two methods shows large values in particular for low wind

speeds at both volcanoes. The difference between the two volcanoes might be due to the fact that the stations at Nevado del Ruiz are placed on the flanks of the volcano, at higher altitude and closer to the crater. The enhanced activity and low wind speeds contribute to the occurrence of wide plumes covering all viewing directions of the scanners.

We interpret scans with a large relative ratio between the two methods as situations, where the recorded spectra show signatures of volcanic $SO_2$ for all viewing directions. Removing an offset value (as for Method A) leads to lower $SO_2$ CDs

and therefore smaller fluxes. It is one important question to ask for the reason for signatures of volcanic $SO_2$ in all viewing directions and what the effect on the $SO_2$ emission rate retrieval is. There are at least two possible explanations for this phenomenon:

1. Radiative transfer effects, i.e. there is no $SO_2$ present in the column defined by the instrument viewing direction, however, a fraction of the radiation passed the plume (and thus picked up an $SO_2$ absorption signature) and then it is scattered into

the instrument's FOV.

2. There is actually $SO_2$ in the instrument FOV.

Model calculations to investigate the influence of radiative transfer on $SO_2$ emission rates at volcanoes were made by Bigge (2015). These radiative transfer model (RTM) calculations indicate, that it is possible to obtain an $SO_2$ signal in viewing directions that should be gas free according to a geometric approach. This means that radiation passes the volcanic plume

before being scattered into the field of view of the spectrometer from a direction that should be gas free. The results of Bigge (2015) showed that the magnitude of these signal depends on the measurement geometry (distance plume-instrument, SZA, extent of the volcanic plume). At Nevado del Ruiz the situation gets further complicated, since clouds are present at the volcano almost throughout the entire year. If the difference between Methods A and B is caused by a radiative transfer effect, it is difficult to judge which one of the methods leads to more accurate results.

For the second explanation, actual $SO_2$ that is apparent for all possible viewing directions of the instrument we have to distinguish further between a broadly dispersed (and moving) plume or $SO_2$ that sits around the instrument without actually moving. The case of an actually broadly dispersed volcanic plume describes a situation in which the volcanic gas plume disperses after being released from the volcano (e.g. due to low wind speeds and thus more time to disperse). In this situation Method A would lead to an underestimation of the $SO_2$ emission rate, while Method B would give a more accurate picture.

However, it is difficult to obtain a an $SO_2$ emission rate from the $SO_2$ column densities in case of a dispersed plume with the current integration schemes used in NOVAC since the plume cross-section cannot be defined accurately in this situation. Another possibility is that volcanic $SO_2$ hovers in the vicinity of the instrument (without actually moving). In this (less likely) case Method A would still lead to accurate $SO_2$ emission rates. Method B would fail to give us a reasonable emission flux since it would add $SO_2$ that is just sitting around the instrument to the real $SO_2$ emission rate.



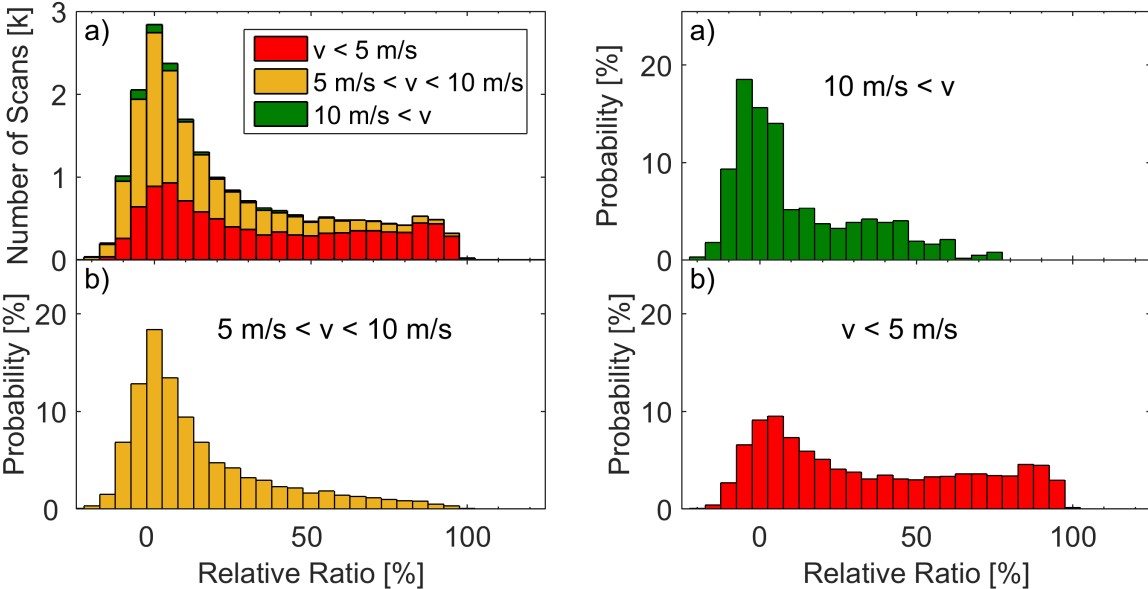

**Figure 15.** Histograms of the relative ratio between the $SO_2$ CDs derived by Method A and Method B for different wind speeds at Nevado del Ruiz. The different histograms show the result for both instruments. In the top left the results for all spectra are shown. The top right shows only spectra where the wind speed was above 10 m/s. The results for lower wind speeds are shown in the lower part of this figure.

Finally there is the possibility that the $SO_2$ contamination originates from background $SO_2$ due to e.g. air pollution or other nearby volcanoes. Also in this case Method A would give the more precise result for the $SO_2$ flux of the volcano under consideration. In summary, however, we believe that a dispersed plume may be the most likely cause for FRS contamination - at least in the cases we investigated - and that the results of Method B give results which are closer to the true $SO_2$ column density than those of Method A. However, there may be situations, where Method A provides more correct data. In any case we recommend to perform both evaluations in order to have a warning for contaminated FRS.

**Appendix A: Calculation of the time offset at Nevado del Ruiz**

The instruments at Nevado del Ruiz had minor problems with the GPS antennas, which led to occasionally wrong time stamps in the spectra during complete days. Under the assumption of a constant daily time offset it is possible to identify time offsets with help of a Langley plot of the $O_3$ CDs. $O_3$ is mainly distributed in the stratosphere and the light-path through the stratosphere largely depends on the SZA. The so-called Air Mass Factor (AMF) can be used to compare the $O_3$ slant column density





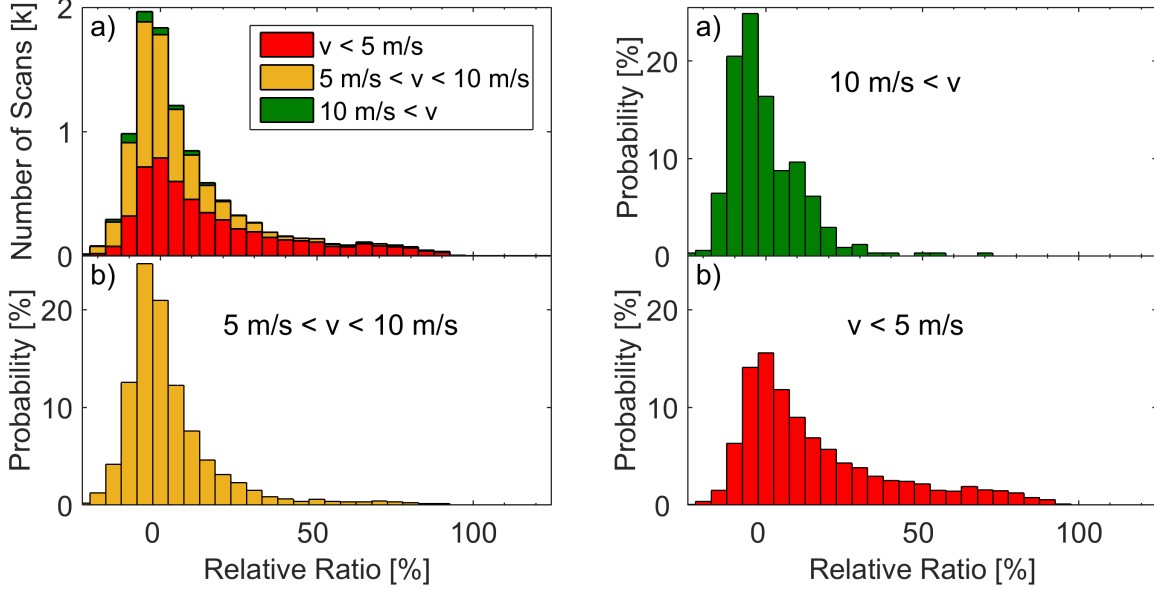

**Figure 16.** Histograms of the underestimation at different wind speeds at Tungurahua.

(SCD) with the vertical column density (VCD). For small values of the SZA $\vartheta$ below $75°$ the AMF can be approximated by (Platt and Stutz, 2008):

$$\text{AMF} = \frac{1}{\cos(\vartheta)} \tag{A1}$$

Since the diurnal $O_3$ VCD variation is small compared to the influence of the SZA, the $O_3$ SCDs for the morning and
5    evening with the same SZA should be similar. We used this property to determine the time of the instrument. We used the $O_3$
CD of the $O_3$ cross-section recorded at $221\,\text{K}$ from Method B and investigated the Langley plot (a plot of the $O_3$ SCD as a
function of AMF). For correct time settings we observed a smooth line with slight curvature, while for incorrect time settings
the Langley plot is not a smooth curve, but shows two distinguishable branches for the morning and afternoon.

In order to calculate the time-shift we applied a time offset and recalculated the SZAs according to Reda and Andreas
10    (2004)[1]. For each offset value a polynomial of $2^{nd}$ order was fitted to the Langley plot and the time offset that maximised $R^2$
was used for this day.



**Table 1.** Table showing data for the instruments at Nevado del Ruiz and Tungurahua. (a) shows the retrieved SO$_2$ CDs from Method B, their variation and how many gas free spectra were taken into account for this statistic. (b) shows the results of a linear fit when plotting Method B - Offset vs. Method A (see Figure 8.) (c) compares the SO$_2$ DOAS retrieval errors and (d) shows the locations of the instrument and statistics on how frequently contaminated spectra exist.

|  |  | Nevado del Ruiz | | Tungurahua | | |
|---|---|---|---|---|---|---|
| | Volcano | | | | | |
| | Station | Bruma | Alfrombrales | Pillate | Bayushig | Huayarapata |
| | Serial Number | D2J2200 | D2J2201 | D2J2140 | I2J8546 | I2J8548 |
| (a) Gas Free Spectra | mean SO$_2$ CD $S$ [molecules/cm$^2$] | $1.0 \cdot 10^{15}$ | $1.4 \cdot 10^{15}$ | $-1.8 \cdot 10^{15}$ | $1.4 \cdot 10^{15}$ | $-6.9 \cdot 10^{15}$ |
| | $\sigma(S)$ [molecules/cm$^2$] | $3.6 \cdot 10^{16}$ | $2.7 \cdot 10^{16}$ | $3.2 \cdot 10^{16}$ | $3.6 \cdot 10^{16}$ | $3.5 \cdot 10^{16}$ |
| | Number of Spectra | $3.3 \cdot 10^{5}$ | $3.5 \cdot 10^{5}$ | $1.8 \cdot 10^{6}$ | $1.6 \cdot 10^{6}$ | $2.5 \cdot 10^{6}$ |
| (b) Plot: SO$_2$ CD (B - Offset) vs. SO$_2$ CD A | Slope | 1.14 | 0.95 | 0.91 | 0.95 | 0.88 |
| | Offset [molecules/cm$^2$] | $-3.0 \cdot 10^{15}$ | $5.2 \cdot 10^{15}$ | $7.7 \cdot 10^{15}$ | $6.8 \cdot 10^{15}$ | $8.7 \cdot 10^{15}$ |
| (c) Mean SO$_2$ Fit Error | Method A [molecules/cm$^2$] | $1.43 \cdot 10^{16}$ | $1.38 \cdot 10^{16}$ | $1.81 \cdot 10^{16}$ | $1.89 \cdot 10^{16}$ | $1.89 \cdot 10^{16}$ |
| | Method B [molecules/cm$^2$] | $1.55 \cdot 10^{16}$ | $1.46 \cdot 10^{16}$ | $1.84 \cdot 10^{16}$ | $1.94 \cdot 10^{16}$ | $1.85 \cdot 10^{16}$ |
| (d) Statistics | Distance from Crater [m] | 3100 | 4150 | 8000 | 11200 | 9400 |
| | Altitude [m. a.s.l.] | 4865 | 4500 | 2550 | 2750 | 2850 |
| | Scanner Geometry | flat | conical | flat | conical | conical |
| | Time frame covered [mm/yy] | 01/10 - 06/12 | 01/09 - 12/11 | 01/09 - 12/11 | 01/09 - 11/10 | 01/09 - 11/11 |
| | Number of Scans | 12826 | 7935 | 4442 | 2331 | 3538 |
| | Relative Ratio $\geq 50\%$ [%] | 22.4 | 20.0 | 4.2 | 7.0 | 10.0 |





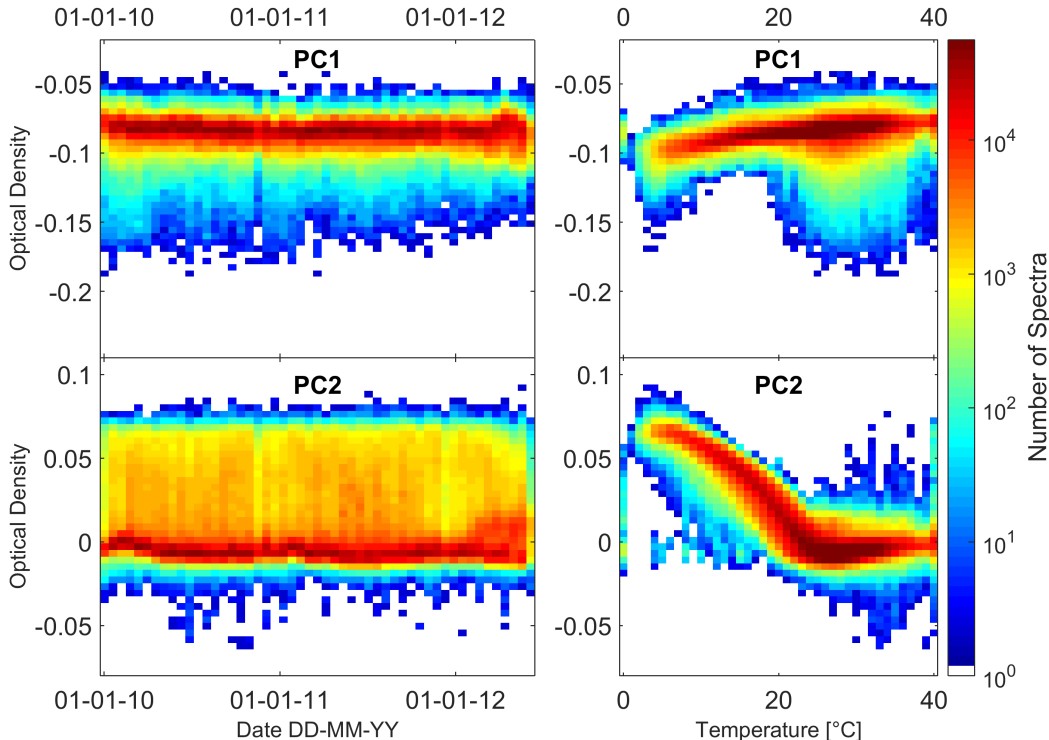

**Figure 17.** Peak-to-peak apparent optical density (e.g. peak-to-peak optical depth of the principal component times the fit coefficient) of the first and second principal components for instrument D2J2200 at Nevado del Ruiz.

## Appendix B: Variation of the principal components

*Acknowledgements.* We would like to thank the European Commission Framework 6 Research Program for funding of the NOVAC project. We kindly acknowledge the NOVAC partners from the Colombian Geological Survey (formerly INGEOMINAS), especially the FISQUIM Research Group and the technical staff at the Manizales Volcanological Observatory as well as the staff at Observatorio del Volcán Tungurahua (IGEPN) for keeping the instruments running for almost a decade at Nevado del Ruiz and Tungurahua. N. Bobrowski thanks for financial support from DFG BO 3611/1-1 and the VAMOS project. The authors thank the DFG project DFG PL193/14-1 for financial support. PL would like to thank Vincent Roy for providing the MatLab script for calculating the Sun's position.

---

[1] We used a MatLab implementation of the Reda and Andreas (2004) algorithm by Vincent Roy.



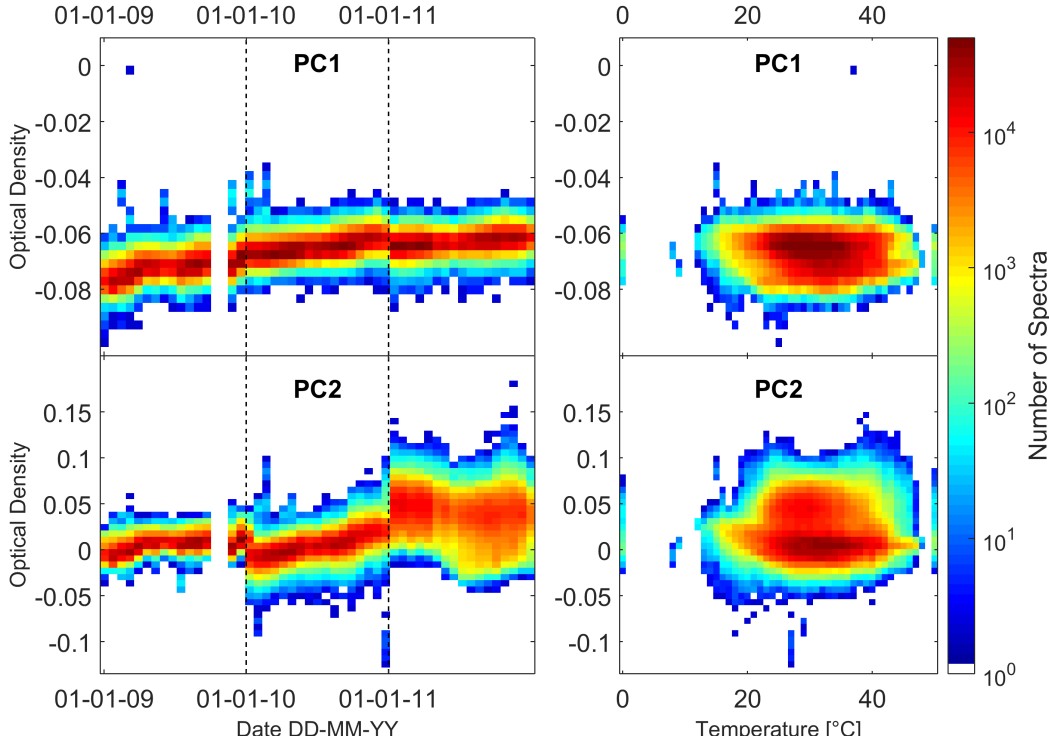

**Figure 18.** Peak-to-peak apparent optical density (e.g. peak-to-peak optical depth of the principal component times the fit coefficient) of the first and second principal components for instrument D2J2140 at Tungurahua. The vertical dashed lines indicate the start of a new time intervals for which a new set of principal components was calculated.

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





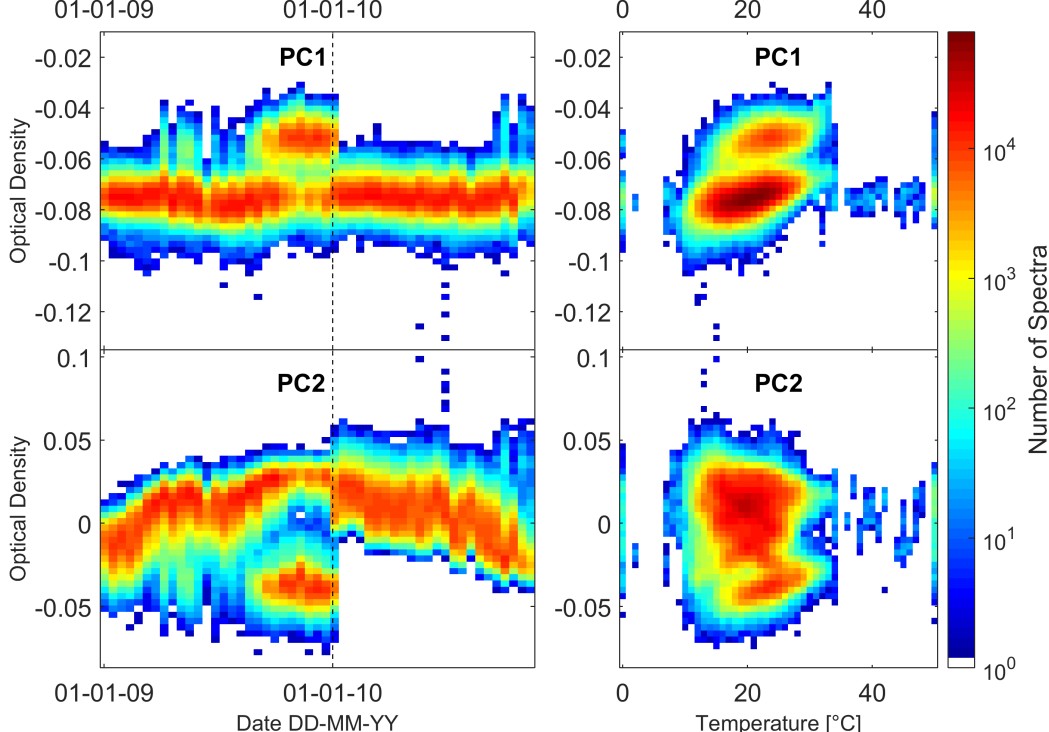

**Figure 19.** Peak-to-peak apparent optical density (e.g. peak-to-peak optical depth of the principal component times the fit coefficient) of the first and second principal components for instrument I2J8546 at Tungurahua. The two distinct values at the end of 2009 might be due to calibration issues due to a drift of the instrument calibration. Creating the principal components more frequently might help in such cases as can be observed at the beginning of 2010.

Burton, M. R. and Sawyer, G. M.: iFit: An intensity-based retrieval for SO2 and BrO from scattered sunlight ultraviolet volcanic plume absorption spectra, Atmospheric Measurement Techniques Discussions, 2016, 1–47, doi:10.5194/amt-2015-380, http://www. atmos-meas-tech-discuss.net/amt-2015-380/, 2016.

Businger, S., Huff, R., Pattantyus, A., Horton, K., Sutton, A. J., Elias, T., and Cherubini, T.: Observing and Forecasting Vog Dispersion 5    from Kilauea Volcano, Hawaii, Bull. Amer. Meteor. Soc., 96, 1667–1686, doi:10.1175/BAMS-D-14-00150.1, http://dx.doi.org/10.1175/ BAMS-D-14-00150.1, 2015.

Bussemer, M.: Der Ring-Effekt: Ursachen und Einfluß auf die spektroskopische Messung stratosphärischer Spurenstoffe, Diplomarbeit, University of Heidelberg, 1993.

Chance, K. and Kurucz, R. L.: An improved high-resolution solar reference spectrum for earth's atmosphere measurements in the ultraviolet, 10    visible, and near infrared, Journal of Quantitative Spectroscopy and Radiative Transfer, 111, 1289–1295, doi:10.1016/j.jqsrt.2010.01.036, http://www.sciencedirect.com/science/article/pii/S0022407310000610, 2010.



Edmonds, M., Herd, R., Galle, B., and Oppenheimer, C.: Automated, high time-resolution measurements of SO2 flux at Soufrière Hills Volcano, Montserrat, Bulletin of Volcanology, 65, 578–586, doi:10.1007/s00445-003-0286-x, http://dx.doi.org/10.1007/s00445-003-0286-x, 2003.

Ferlemann, F.: Ballongestützte Messung stratosphärischer Spurengase mit differentieller optischer Absorptionsspektroskopie, Ph.D. thesis, Heidelberg, Univ., Diss., 1998, 1998.

Galle, B., Oppenheimer, C., Geyer, A., McGonigle, A. J. S., Edmonds, M., and Horrocks, L.: A miniaturised ultraviolet spectrometer for remote sensing of SO$_2$ fluxes: a new tool for volcano surveillance, Journal of Volcanology and Geothermal Research, 119, 241–254, doi:10.1016/S0377-0273(02)00356-6, http://www.sciencedirect.com/science/article/pii/S0377027302003566, 2003.

Galle, B., Johansson, M., Rivera, C., Zhang, Y., Kihlman, M., Kern, C., Lehmann, T., Platt, U., Arellano, S., and Hidalgo, S.: Network for Observation of Volcanic and Atmospheric Change (NOVAC) - A global network for volcanic gas monitoring: Network layout and instrument description, Journal of Geophysical Research: Atmospheres, 115, n/a–n/a, doi:10.1029/2009JD011823, http://dx.doi.org/10.1029/2009JD011823, 2010.

Grainger, J. and Ring, J.: Anomalous Fraunhofer Line Profiles, nature, 193, 762, 1962.

Hastie, T., Tibshirani, R., and Friedman, J.: The elements of statistical learning, vol. 1, Springer New York, http://statweb.stanford.edu/~tibs/ElemStatLearn/, 2001.

Kraus, S. G.: DOASIS - A Framework Design for DOAS, Ph.D. thesis, University of Mannheim, 2006.

Lampel, J.: Measurements of reactive trace gases in the marine boundary layer using novel DOAS methods, Ph.D. thesis, University of Heidelberg, 2014.

Li, C., Joiner, J., Krotkov, N. A., and Bhartia, P. K.: A fast and sensitive new satellite SO2 retrieval algorithm based on principal component analysis: Application to the ozone monitoring instrument, Geophysical Research Letters, 40, 6314–6318, doi:10.1002/2013GL058134, http://dx.doi.org/10.1002/2013GL058134, 2013.

Lübcke, P.: Optical remote sensing measurements of bromine and sulphur emissions, Ph.D. thesis, Heidelberg, Univ., Diss., 2014, http://www.ub.uni-heidelberg.de/archiv/16879, zsfassung in dt. Sprache, 2014.

Lübcke, P., Bobrowski, N., Arellano, S., Galle, B., Garzón, G., Vogel, L., and Platt, U.: BrO/SO$_2$ molar ratios from scanning DOAS measurements in the NOVAC network, Solid Earth Discussions, 5, 1845–1870, doi:10.5194/sed-5-1845-2013, http://www.solid-earth-discuss.net/5/1845/2013/, 2013.

McGonigle, A. J. S., Inguaggiato, S., Aiuppa, A., Hayes, A. R., and Oppenheimer, C.: Accurate measurement of volcanic SO$_2$ flux: Determination of plume transport speed and integrated SO$_2$ concentration with a single device, Geochemistry, Geophysics, Geosystems, 6, n/a–n/a, doi:10.1029/2004GC000845, http://dx.doi.org/10.1029/2004GC000845, 2005.

Moffat, A. J. and Millan, M. M.: The applications of optical correlation techniques to the remote sensing of SO$_2$ plumes using sky light, Atmospheric Environment, 5, 677–690, doi:10.1016/0004-6981(71)90125-9, http://www.sciencedirect.com/science/article/pii/0004698171901259, 1971.

Mori, T. and Burton, M.: The SO$_2$ camera: A simple, fast and cheap method for ground-based imaging of SO$_2$ in volcanic plumes, Geophysical Research Letters, 33, n/a–n/a, doi:10.1029/2006GL027916, http://dx.doi.org/10.1029/2006GL027916, 2006.

Pearson, K.: LIII. On lines and planes of closest fit to systems of points in space, The London, Edinburgh, and Dublin Philosophical Magazine and Journal of Science, 2, 559–572, 1901.

Perner, D. and Platt, U.: Detection of nitrous acid in the atmosphere by differential optical absorption, Geophysical Research Letters, 6, 917–920, doi:10.1029/GL006i012p00917, http://dx.doi.org/10.1029/GL006i012p00917, 1979.



Pinardi, G., Roozendael, M. V., and Fayt, C.: The influence of spectrometer temperature variability on the data retrieval of SO$_2$., In NOVAC second annual activity report, pp 44–48, NOVAC consortium, 2007.

Platt, U. and Stutz, J.: Differential Optical Absorption Spectroscopy - Principles and Applications, Physics of Earth and Space Environments, Springer Berlin Heidelberg, 2008.

Platt, U., Marquard, L., Wagner, T., and Perner, D.: Corrections for zenith scattered light DOAS, Geophysical Research Letters, 24, 1759–1762, doi:10.1029/97GL01693, http://dx.doi.org/10.1029/97GL01693, 1997.

Reda, I. and Andreas, A.: Solar position algorithm for solar radiation applications, Solar Energy, 76, 577 – 589, doi:http://dx.doi.org/10.1016/j.solener.2003.12.003, http://www.sciencedirect.com/science/article/pii/S0038092X0300450X, 2004.

Salerno, G., Burton, M., Oppenheimer, C., Caltabiano, T., Tsanev, V., and Bruno, N.: Novel retrieval of volcanic SO$_2$ abundance from ultraviolet spectra, Journal of Volcanology and Geothermal Research, 181, 141–153, doi:10.1016/j.jvolgeores.2009.01.009, http://www.sciencedirect.com/science/article/pii/S0377027309000158, 2009.

Shefov, N. N.: Intensivnosti nokotorykh emissiy sumerochnogo i nochnogo neba (Intensities of some Emissions of the Twilight and Night Sky), Spectral, electrophotometrical and radar researches of aurorae and airglow, IGY program, section IV, 1, 25, 1959.

Smith, L. I.: A tutorial on principal components analysis, Cornell University, USA, 51, 52, 2002.

Stoiber, R. E., Malinconico, L. L., and Williams, S.: Use of the correlation spectrometer at volcanoes, in: Forecasting volcanic events, edited by Tazieff, H. and Sabroux, J.-C., pp. 424–444, Elsevier Science Pub. Co., Inc., New York, NY, 1983.

Vandaele, A., Hermans, C., and Fally, S.: Fourier transform measurements of SO$_2$ absorption cross sections: II.: Temperature dependence in the 29000 - 44000 cm$^{-1}$ (227-345 nm) region, Journal of Quantitative Spectroscopy and Radiative Transfer, 110, 2115–2126, doi:10.1016/j.jqsrt.2009.05.006, http://www.sciencedirect.com/science/article/pii/S0022407309001800, 2009.

Wenig, M., Jähne, B., and Platt, U.: Operator representation as a new differential optical absorption spectroscopy formalism, Applied optics, 44, 3246–3253, 2005.