# Peer review of "Retrieval of absolute $SO_2$ column amounts from scattered-light spectra - Implications for the evaluation of data from automated DOAS Networks."

_Atmospheric Measurement Techniques, 2016_

## Referee Comment (RC1) · Anonymous Referee #1 · 10 May 2016

The manuscript addresses a relevant source of uncertainty in the determination of SO2 fluxes by scanning DOAS instruments that might occur at many monitored volcanoes during certain wind conditions and depending on the geometrical parameters of the scanning instruments. When no plume free region is within the instrument's range, the use of a Fraunhofer reference spectrum from the same scan produces erroneous results with respect to the absolute column densities and the authors propose the use of a modelled reference from a high resolution solar spectrum. The DOAS retrieval error, which normally increases by the straight forward use of a modelled Fraunhofer reference spectrum instead of a measured one, is greatly reduced by the principal component analysis that the authors apply. The manuscript is well written and clearly

structured and should, in my opinion, be accepted as is.

---

## Referee Comment (RC2) · A.nbsp;J.nbsp;S. McGonigle (Referee) · 10 May 2016

Thank you for the opportunity to review for AMT.

I am recommending minor revisions here.

This article concerns an important area of interest in terms of scanning DOAS measurements of volcanic sulphur dioxide emissions. These data are now routinely collected at a number of volcanoes worldwide, in particular via the NOVAC project, which this article pertains to. The data are a significant component of the work of volcano observatories in terms of monitoring and this article describes an important contribution to ensuring data integrity in respect of spectral processing methodologies. In particular the paper details an approach concerning use of modeled rather than measured Fraunhofer spectra in the retrievals.

In my opinion the paper is very well written with excellent structure throughout, strong demonstration of the key points and scientific integrity.

The only point which I would recommend that the authors consider, is regarding the operational utility of this approach. There is no doubt that could be a useful refinement of currently adopted protocols in terms of scanning DOAS measurements. However, the transference of this approach into routine application in volcano observatories internationally, in particular by staff who are not necessarily expert in remote sensing, could be another matter. Perhaps the authors could comment on how transferable this approach could be into operational utilisation e.g., would it require spectroscopic knowledge from the user community, or could user-friendly programs be written to enable non-experts to benefit from these developments. Some more text anticipating how this approach could be transitioned into operational usage could really help with uptake from the end user community.

---

## Referee Comment (RC3) · Anonymous Referee #3 · 13 May 2016

Review of Manuscript AMT-2016-24: Retrieval of absolute SO2 column amounts from scattered-light spectra - Implications for the evaluation of data from automated DOAS Networks by P. Lübcke et al.,

Recommendations I recommend publication in AMT only after a MAJOR REVISION of the manuscript in line with criteria requirements of the Journal.

General comments This paper discuss the implementation of a modelled Fraunhofer Reference Spectrum (FRS) for retrieving SO2 Column Density (CD) from open-path ultraviolet spectra by scanning ultraviolet spectrometer. The authors carry out a number of specific statistical analysis (e.g., PCA) such as to identified instrumental features of spectrometers and explore confidence in the retrieval. Data from NOVAC network at

[Figure]

Nevado del Ruiz (Colombia) and Tungurahua (Ecuador) between 2010 and 2012, and 2009 and 2011, respectively were reduced applying both the standard NOVAC (e.g., Galle et al., 2010) and the FRS approach. Comparing the results gathered from both methods, the authors observed a large difference between the two, with the NOVAC underestimating SO2 CDs more than a factor of two. At volcanology observatories self-reliance from robotic system is a first priority for real-time multidisciplinary surveillance and monitoring purpose. NOVAC is largely spread at active volcanoes worldwide representing a unique trail for the understanding of volcanoes and their gas emission impact in climatology. Hence, the application of the modelled FRS at NOVAC represents a step forward for both monitoring purpose and revaluation of global volcanic emission rates inventory.

I believe this paper is in line with the scope of AMT. However, there are some aspects of the work that are questionable, and in which a strong revision is required before the manuscript is suitability for publication in AMT. There are also several minor points to address, which I have described with line references after my general comments.

The main issue concern the originality of this study. This research is not a novel idea in SO2 measurements at active volcanoes by scanning spectrometer system. In particular, the pioneering application of a modelled FRS in NOVAC data at the Piton de la Fournaise has be not mentioned and/or discussed (Hibert et al., 2015). In this work, the authors successfully retrieved SO2 CD and flux from the NOVAC scanning network without invoke different ILS as in the manner of Salerno et al., 2009a and no laborious PCA analysis for accounting of instrumental effects in the spectrometers. Nevertheless, Lubcke et al., set the originality of their study in the PCA analysis. The authors carried out this statistical test in free-volcanic gas spectra selected from the database of Nevado del Ruiz and Tunguraha NOVAC network. Selected 7-days data between Sept-Oct at Nevado del Ruiz and 10-days/year between 2009 and 2011 at Tunguraha. Results, were assumed reflecting the instrumental features of the entire network. However, as reported by Pinardi et al., 2007, instrumental effect dramatically

may change over time due to temperature drift, and these are different between spectrometers (eg., Instrumental line function, stray lights). As the NOVAC instruments are not thermo-stabilised (Galle et al., 2010) variable instrumental drifts may take place at different magnitude and time scale. This raises the hesitation to believe that the PCA results gathered from few data and from a unique spectrometer (per network) may be assumed representative of the instrumental features of entire scanning networks. The authors, attempt to explore this issue by comparing the residual structures and optical depth (Fig. 3, 4, and 5) obtained by retrieving the entire records of data from the two networks. They found a reasonable agreement between the instruments. This find arise anyway a further question. In detail, since the results of the comparison relates to the PCA results and as instrumental drifts are taking place physically in different way, the question that come up is whether the statistical PCA has a fundamental role in the framework of the paper.

There are also some comments about the structures of the article. The paper needs to be restructured, reducing the number of paragraphs and length of the article itself. For instance, the detailed description of the DOAS technique is unnecessary. This technique is a very well established spectroscopy approach and the theory largely published. I really suggest the authors to replace details and equation with references (Platt and Stutz, 2008). Moreover, some paragraph might be eliminated or merged with others, e.g 4.2 consist of only five lines. Instead, I'd suggest the authors to include a small paragraph describing the NOVAC networks object of the study. Finally, a graph of the SO2 CDs retrieved by FRS throughout the years at both Nevado del ruiz and Tunguraha should be displayed for completeness of the study.

Other comments Abstract: Pag 1, Section 05, line 1: define here the scanning DOAS, change this sentence with '.........scanning ultraviolet spectrometers network, also named as scanning DOAS '. Pag 1, Section 10, line 3: specify volcanoes. Pag 1, section 20, line 1: 'complicated instrumental calibrations' please report which kind of calibrations are required in field 'eg.,.....' . Pag 2, section 05, line 3-4: It's not clear if

the word 'New', is related to the FRS or the entire study carried out in the paper. In case it relates to the FRS method, as mentioned in the general comment, I'd suggest to remove 'new' because it is not a novel approach.

Introduction Pag 2, section 10, line 2: add reference Williams-Jones, et al, 2008 and Oppenheimer, 2011 Pag 2, section 10, line 2-3: More recently the availability of miniature spectrometers allowed. . . . . ...': I suggest to delete more recently because it's now more than twelve years after the first application of this technique at active volcanoes by Galle et al., 2003. Pag 2, section 10, line 5: add reference 'Elias et al, 2006. Pag 2, section 15, line 2-3: change 'One of the first installations were scanning DOAS instruments at Montserrat volcano (Edmonds et al., 2003)' with 'The first installations of scanning-DOAS network was developed at Montserrat volcano (Edmonds et al., 2003)'. Pag 2, section 20, line 1: add reference Salerno et al., 2009b. Pag 3, section 15, line 1-4: specify large NOVAC database of the two Colombian and Ecuadorian volcanoes object of this research. Pag 3, section 30, line 5-7: move and change 'First steps towards the here described approach were taken in Lübcke (2014), where measurements from NOVAC instruments at Nevado del Ruiz were evaluated for SO2 with a modelled background spectrum'. With 'this approach was successively adopted by Lübcke (2014) and Hibert et al., 2015 for evaluating NOVAC data collected at Nevado del Ruiz and at Piton de la Fournaise Reunion, respectively'. Pag 3, section 10-15, line 3-3: move and rephrase this section in Pag. 2 section 25-30 while talking about the use of the modelled FRS. Report that limitations and uncertainty of the standard NOVAC retrieval has been already discussed and overtaken applying a modelled background FRS by Hibert, C., et al. 2015.

2 Background spectra for scanning DOAS instrument networks at volcanoes

As reported in the general comments, the theory behind the DOAS evaluation is widely published. Thus allocate paragraph to a detailed description of the physics/equation of DOAS does not provide any advantages to the article. I'd suggest to replace part of this section with Platt and Stutz, 2008. Pag 4, section 30, line 7-8: The experiment

reported in Salerno et al., 2009a does not imply any routinely specific spectroscopy retrieval operations to be performed in field. SO2 calibrated quartz cells spectra were employed for exploring and validate at different timescale the application of a modelled FRS for reducing ultraviolet open-path spectra from scanning ultraviolet spectrometers. The comparison of three year of SO2 CDs an flux form scanning system and traverses (Salerno et al., 2009b) provide a further quantitative constrain on the efficiency of the approach developed in Salerno et al., 2009a. Pag 5, section 10, line 1-5: rephrase this section and delete math equations. Pag 5, section 15, line 4-5: As reported in the paper (pag 6, section 20) SO2 retrieval was performed between 310.0 - 326.8 nm, report the reason for which high resolution laboratory cross-sections were convolved using the Hg line at 334.15 nm and not for instance 313.16 nm.

3 Data evaluation 3.1 Settings of the DOAS retrieval

Pag 6, section 15, line 5: 'for both retrieval methods (see below)' define here the two retrievals. Pag 6, section 25-30, line 4-9: detail of standard NOVAC retrieval has been already reported in the introduction, delete this section or synthesise. Pag 7-8, section 30 – 30: replace DOAS theory with references.

3.3 Principal Component Analysis for Method B Pag 9, section 10-20, the authors report that PCA neds to be run using clean-gas spectra. However, due to volcanic activity this is not so straightforward. Therefore, only few dataset of spectra were selected from both Nevado del Ruiz and Tungurahua database. Nevertheless, at Pag 14 paragraph 4.3 the requirements of clean-gas spectra arise again for evaluating the sensitivity of the retrieval as zero CDs. As reported in this paragraph at section 5, line 2-4 a greater number of data-days were discriminated at both volcanoes respect what have been selected for the PCA analysis (e.g, at Nevado del Ruiz 7 days vs 73 days). Please, rephrase the two section in case of mistake or clarify this conflict issue. Pag 9, section 15, line 5: constrain in a quantitative scale the meaning of 'little degassing activity'. Pag 9-10, section 30-5, define period of selected data at Tungurahua. Pag 10, section 30, line 7: constrain in a quantitative scale the meaning of 'low volcanic degassing activity'.

3.4 Implementation of the new algorithm

I would suggest the authors to remove this paragraph. It is a summary of what has been already reported in the manuscript.

4 Results

Pag 11, section 25, line 1: delete brackets in Lubcke et al., 2013. Pag 14, section 05: see comments Pag 9, section 10-20. Conflict or erroneous explanation of this issue. Pag 17-18, section 05. Method A and B retrieve the same spectra, explain in more detail why the effect of stray light should affect the two retrieval performed with method A and B in a different way.

4.2 DOAS fit example This paragraph consist of only five lines, please remove it or include in a different paragraph.

Figure Fig 9: change 'which is similar to the standard NOVAC approach' with 'standard NOVAC approach'.

Litterature Elias, T., Sutton, A.J., Oppenheimer, C., Horton, K.A., Garbeil, H., Tsanev, V., McGonigle, A. J.S., Williams-Jones, G., 2006. Intercomparison of COSPEC and two miniature ultraviolet spectrometer systems for SO2 measurements using scattered sunlight. Bull. Volcanol. 68, 313–322.

Platt, U. and Stutz, J.: Differential Optical Absorption Spectroscopy - Principles and Applications, Physics of Earth and Space Environments, Springer Berlin Heidelberg, 2008.

Galle, B., M. Johansson, C. Rivera, Y. Zhang, M. Kihlman, C. Kern, T. Lehmann, U. Platt, S. Arellano, and S. Hidalgo (2010). Network for Observation of Volcanic and Atmospheric Change (NOVAC)—A global network for volcanic gas monitoring: Network layout and instrument description, J. Geophys. Res., 115, D05304, doi:10.1029/2009JD011823.

Galle, B., Oppenheimer, C., Geyer, A., McGonigle, A.J.S., Edmonds, M., Horrocks, L.A., 2003. A miniaturised ultraviolet spectrometer for remote sensing of SO2 fluxes: a new tool for volcano surveillance. J. Volcanol. Geotherm. Res. 119, 241–254.

Lübcke, P., et al., 2015, DOAS evaluation of volcanic SO2 using a modeled background spectrum: Examples from the NOVAC stations at Nevado del Ruiz (Colombia) and Tungurahua (Ecuador), Geophysical Research Abstracts Vol. 17, EGU2015-1803-1, EGU General Assembly 2015, Vienna.

Hibert, C., A. Mangeney, M. Polacci, A. D. Muro, S. Vergniolle, V. Ferrazzini, A. Peltier, B. Taisne, M. Burton, T. Dewez, G. Grandjean, A. Dupont, T. Staudacher, F. Brenguier, P. Kowalski, P. Boissier, P. Catherine, and F. Lauret (2015), Toward continuous quantification of lava extrusion rate: Results from the multidisciplinary analysis of the 2 January 2010 eruption of Piton de la Fournaise volcano, La Réunion. J. Geophys. Res. Solid Earth, 120, 3026–3047. doi: 10.1002/2014JB011769.

Oppenheimer, 2011, Ultraviolet sensing of volcanic sulfur emissions, Elements, Vol. 6, pp. 87–92, DOI: 10.2113/gselements.6.2.87.

Salerno, G. G., M. R. Burton, C. Oppenheimer, T. Caltabiano, D. Randazzo, N. Bruno, and V. Longo (2009b), Three-years of SO2 flux measurements of Mt. Etna using an automated UV scanner array: Comparison with conventional traverses and uncertainties in flux retrieval, J. Volcanol. Geotherm. Res., 183, 76–83, doi:10.1016/j.jvolgeores.2009.02.013.

Williams-Jones, G., Stix, J., Hickson, C., 2008. The COSPEC Cookbook: making SO2 measurements at active volcanoes. IAVCEI, Methods in Volcanology, vol. 1. and Oppenheimer, 2011, Ultraviolet sensing of volcanic sulfur emissions, Elements, Vol. 6, pp. 87–92, DOI: 10.2113/gselements.6.2.87.

---

## Referee Comment (RC4) · Anonymous Referee #4 · 6 Jun 2016

This paper presents an approach to analyse UV scattered sunlight spectra of volcanic plumes, such that an absolute rather than relative quantification of SO2 slant column amounts can be achieved. The motivation for this approach is that the NOVAC network of scanning UV spectrometers has used since its inception an approach where a reference spectrum from within each scan is used to remove solar spectrum features. SO2 values are then offset by the minimum SO2 value in each scan. This has the advantage of excellent solar feature removal, but also means that if SO2 is ubiquitous then the minimum SO2 value will not represent zero SO2, and an underestimate of column amounts will affect the scan, leading to flux underestimates. On the contrary, by using an absolute method for SO2 retrieval this problem is addressed.

[Figure]

The approach proposed is effectively identical to that reported in Salerno et al. 2009a, which has been applied successfully by INGV Italy to the scanning UV spectrometer SO2 flux monitoring networks on Etna and Stromboli since 2006. Salerno 2009a used a high resolution UV solar spectrum (Kurucz et al., 1984) to model the Fraunhofer features of the solar spectrum, whereas Lubcke et al. use the Chance and Kurucz (2010) Solar atlas.

Salerno et al., 2009b examined three years' results using the algorithm described in Salerno et al., 2009a from the automated flux monitoring network on Mt. Etna with traverse flux measurements, which are in many ways much more robust than scanning flux measurements: a clear sky spectrum can be collected which is guaranteed to be SO2 free, and there are no geometric corrections to make for plume height.

Burton and Sawyer 2016 present an update to the Salerno et al. 2009a work in which they recognise that there are instrumental response features which add a fixed pattern to the measured spectra using CCD-based spectrometers, the 'flat' spectrum. This adds 1-3% noise but can be readily characterised and removed from each spectrometer using lab-based measurements of a broad-band UV source.

The issues associated with the standard NOVAC analysis system have been well known for over a decade, since the implementation of the approach of Salerno et al. 2009a. The main objective of the Lubcke paper is that (page 3 line 1-3) "This work will follow the idea of using a high resolution Solar atlas spectrum (Chance and Kurucz, 2010) in order to calculate a gas free background spectrum which is used as an FRS for the DOAS evaluation of SO2.". It appears however that this work has already been done by Salerno et al. 2009, so the novelty of the current work should lie elsewhere. This immediately highlights that a major refocussing of the manuscript is needed, because so much of the work has already been developed.

A valuable contribution of the Lubcke paper is the comparison of the SO2 offset correction with the Salerno et al. 2009a type retrieval for data from Nevada del Ruiz and

[Figure]

Tungurahua. However, with 30 volcanoes in the NOVAC network and the fact that an established method for dealing with the issues of absolute SO2 retrievals has been applied for over a decade, it seems wholly inadequate that only two volcanoes are investigated. My recommendation is therefore to refocus this work from a rehash of previously published methods to a thorough investigation of the full implications of the application of the SO2 offset approach to the NOVAC network. Sincerely, with the time available since this issue was highlighted and a solution shown it is not acceptable to present just for two volcanoes from 30. A further requirement for a revised or re-submitted paper is that some comparison with traverse flux data is used. The final objective of all this work is to get as accurate as possible SO2 flux data, not SO2 slant column amounts. The corrections posed may have an angular dependence, leading to unexpected impacts on the reconstructed SO2 fluxes from scans. The only way to evaluate this properly is to compare scanner SO2 fluxes with traverse flux measure-ments, as performed by Salerno et al., 2009b, which should also be referenced here. I would therefore strongly recommend that such measurements and their comparison be included in a future revision, as otherwise the veracity of the final SO2 flux mea-surements will remain doubtful.

The final contribution of the Lubcke paper is a PCA analysis which effectively does a qualitative job of removal of the flat spectrum as described by Burton & Sawyer 2016. The PCA approach means the user absolves themselves of responsibility for finding the physical process producing the observed features, which is rather disappointing, as precise physical understanding of the measurement process is how we can improve in the future. The majority of the features captured by PCA are flat spectrum, so this contribution is somewhat redundant, given that a clear physical explanation of the pro-cess is given by Burton & Sawyer 2016.

The present work is also overly long with a redundant exhaustive explanation of the basic retrieval technique. Significant shortening is essential.

Burton, M. R. and Sawyer, G. M.: iFit: An intensity-based retrieval for SO2 and

[Figure]

BrO from scattered sunlight ultraviolet volcanic plume absorption spectra, Atmospheric Measurement Techniques Discussions, 2016, 1–47, doi:10.5194/amt-2015-380, http://www.atmos-meas-tech-discuss.net/amt-2015-380/, 2016.

Kurucz, R.L., Furenlid, I., Brault, J.,and Testerman, L., 1984. Solar flux atlas from to 1300 nm, Natl. Sol. Obs., Sunspot, New Mexico, 240.

Salerno, G.G., Burton, M., Oppenheimer, C., Caltabiano, T., Tsanev, V.I., Bruno, N., 2009a, Novel retrieval of volcanic SO2 abundance from ultraviolet spectra. J. Volcanol. Geotherm. Res. 181, 141–153. doi:10.1016/j.jvolgeores.2009.01.009.

Salerno, G.G., Burton, M.R., Oppenheimer, C., Caltabiano, T., Randazzo, D., Bruno, N., Longo, V., 2009b, 183, 76-83. Three-years of SO2 flux measurements of Mt. Etna using an automated UV scanner array: comparison with conventional traverses and uncertainties in flux retrieval, doi:10.1016/j.jvolgeores.2009.02.013

---

## Author Comment (AC1) · 2 Aug 2016

**Dear Editor,**

**We like to thank our colleague for the comments and suggestions. We implemented his suggestions into the revised version of our manuscript. With these changes (and those in response to the other reviewers') we are confident that the revised manuscript has improved considerably. Please find the comments (in normal face) and our answers (in bold face) below:**

Thank you for the opportunity to review for AMT. I am recommending minor revisions here.

**We like to thank Dr. Andrew McGonigle for his review.**

This article concerns an important area of interest in terms of scanning DOAS measurements of volcanic sulphur dioxide emissions. These data are now routinely collected at a number of volcanoes worldwide, in particular via the NOVAC project, which this article pertains to. The data are a significant component of the work of volcano observatories in terms of monitoring and this article describes an important contribution to ensuring data integrity in respect of spectral processing methodologies. In particular the paper details an approach concerning use of modelled rather than measured Fraunhofer spectra in the retrievals.

In my opinion the paper is very well written with excellent structure throughout, strong demonstration of the key points and scientific integrity.

**We appreciate the positive evaluation of our manuscript.**

The only point which I would recommend that the authors consider, is regarding the operational utility of this approach. There is no doubt that could be a useful refinement of currently adopted protocols in terms of scanning DOAS measurements. However, the transference of this approach into routine application in volcano observatories internationally, in particular by staff who are not necessarily expert in remote sensing, could be another matter. Perhaps the authors could comment on how transferable this approach could be into operational utilisation e.g., would it require spectroscopic knowledge from the user community, or could user-friendly programs be written to enable non-experts to benefit from these developments. Some more text anticipating how this approach could be transitioned into operational usage could really help with uptake from the end user community.

**We thank the referee for this excellent suggestions. As he correctly points out, the approach encompasses some tasks which require an end-user with a certain level of knowledge in the evaluation of spectroscopic data, however our approach was developed from the beginning with the aim to ultimately implement it into operational spectroscopic networks. We, therefore, will add the following discussion to the conclusions:**

"The approach presented was developed with the aim to implement it into the operational monitoring of spectroscopic networks. Its great advantage is reduction of manual labour (field measurements and calibration) at the expense of a more elaborate statistical evaluation. The main challenge for the implementation is the availability of a 'training set' of field-spectra, which is guaranteed to be gas free (the importance of such a data set was also suggested in Burton and Sawyer, 2016). Additionally, the end-user has to set a number of instrument specific parameters, which can potentially influence the performance of the retrieval. These parameters are mainly connected to the PCA and include the cut-off value determining which spectra are excluded from PCA, accounting for hot-pixels of the detector (which leave a dominant structure in the residual) and a well chosen number of principal components to include in the DOAS retrieval.

At present, the algorithm can easily be applied to any other volcano of the NOVAC network off-line, but it is not part of the standard software used by the observatories in the network. The main advantage of implementing this method in the future will be the possibility to identify plume scans not having an $SO_2$-free spectrum that gives the baseline zero $SO_2$ level. Currently lack of a $SO_2$-free reference spectrum will result in the standard NOVAC software not calculating the correct $SO_2$ emission rate for such measurements. In these cases, applying our new method yields the important information that the $SO_2$ emission rate from the volcano is non-zero if one of the following conditions prevails: 1) The plume is extending at low altitude towards the spectrometer site, and/or 2) The plume is elevated but sufficiently extended to fill the entire range of scan angles. Even if those cases may only occur rarely at some volcanoes, it is highly advantageous to obtain information on gas emissions in these cases to complement monitored time series. Last but not least, the presented method confirms $SO_2$ emission rate measurements with low or no degassing present because it eliminates the chance that those are resulting from contaminated FRS."

---

## Author Comment (AC2) · 2 Aug 2016

Please find our answers to the comment in the attached pdf file.

Please also note the supplement to this comment:
http://www.atmos-meas-tech-discuss.net/amt-2016-24/amt-2016-24-AC2-supplement.pdf

---

## Author Comment (AC4) · 2 Aug 2016

We like to thank anonymous referee #1 for his review of our manuscript.

---

## Author Comment (AC3)

**Dear Editor,**

**We like to thank referee #4 for the comments and suggestions. We implemented most of the suggestions into the revised version of our manuscript and give detailed reasons in the cases where we could not follow the reviewer's suggestions. With these changes (and those in response to the other reviewers' we are confident that the revised manuscript has improved considerably. Please find the comments (in normal face) and our detailed answers (in bold face) below:**

**Anonymous Referee #4**

This paper presents an approach to analyse UV scattered sunlight spectra of volcanic plumes, such that an absolute rather than relative quantification of SO2 slant column amounts can be achieved. The motivation for this approach is that the NOVAC network of scanning UV spectrometers has used since its inception an approach where a reference spectrum from within each scan is used to remove solar spectrum features. SO2 values are then offset by the minimum SO2 value in each scan. This has the advantage of excellent solar feature removal, but also means that if SO2 is ubiquitous then the minimum SO2 value will not represent zero SO2, and an underestimate of column amounts will affect the scan, leading to flux underestimates. On the contrary, by using an absolute method for SO2 retrieval this problem is addressed.
**We thank the referee for the comments and the good suggestions, however we do not agree to all of the referee's opinions and will give detailed reasons for this below.**

The approach proposed is effectively identical to that reported in Salerno et al. 2009a, which has been applied successfully by INGV Italy to the scanning UV spectrometer SO2 flux monitoring networks on Etna and Stromboli since 2006. Salerno 2009a used a high resolution UV solar spectrum (Kurucz et al., 1984) to model the Fraunhofer features of the solar spectrum, whereas Lubcke et al. use the Chance and Kurucz (2010) Solar atlas.
**We strongly disagree with this criticism. While both methods use a version of the Chance and Kurucz Solar atlas spectrum as a baseline to model the FRS, its implementation and details of retrieval are quite different which makes a large difference in both, implementation as well as applicability of the approaches. Salerno et al. 2009a empirically tuned different evaluation parameters in order to reproduce the SO$_2$ column density of calibration cells with a known column amount. The authors later applied the settings they found for two instruments to three different instruments and found good evaluation performance.**
**However, their approach suffers from some limitations. For example, during the procedure they made choices that are not reasonable from a physics standpoint. The authors used different instrument line functions for the convolution of the FRS, the O$_3$ and the SO$_2$ absorption cross-sections. These errors were most likely made in order to reduce the**

influences of the CCD's quantum efficiency as well as the spectrometers grating, which were not explicitly mentioned in their manuscript.

**It appears to us that the reviewer did not entirely understand the motivation and aim of our work, we need to correct the properties of spectrometers, which are (or were) in the field and are practically inaccessible for characterization measurements. Our approach allows to evaluate these kind of data sets, even from instruments that are not accessible for laboratory characterization. In our opinion, this is a crucial advantage, since most of the spectrometers in the NOVAC network are installed in remote locations that cannot be easily accessed. While we acknowledge that the approaches by Salerno et al., 2009a and Burton and Sawyer 2016 may have worked for the instruments installed at Mt. Etna or for individual car traverse measurements, they are not feasible for re-evaluating already existing data-sets. As was suggested by Referee #3, it cannot be concluded from single instruments, that the approach and the parameters found by Salerno et al., 2009a would work for instruments that are installed at different locations and are subject to a different variation of, e.g., temperature.**

Salerno et al., 2009b examined three years' results using the algorithm described in Salerno et al., 2009a from the automated flux monitoring network on Mt. Etna with traverse flux measurements, which are in many ways much more robust than scanning flux measurements: a clear sky spectrum can be collected which is guaranteed to be SO2 free, and there are no geometric corrections to make for plume height.

**We agree that the work of Salerno et al., 2009b deserves greater credit. We will remedy this shortcoming of our manuscript and cite Salerno et al., 2009b in the introduction as well as the discussion of our results. However, we also notice that trying to validate flux measurements taken by two different instruments and measurement strategies has its own issues (as is discussed in Salerno et al., 2009b as well). The results of scanning and traverse DOAS measurements are expected to agree only if sufficient care is taken on controlling the observation geometry in a way that both instruments sample about the same section of the plume at nearly the same time, and the potential influence of radiative transfer effects caused by different pointing directions and distances to the plume. Thus the comparison is limited by the uncertainty of factors like these that usually escape control. The best way to test a retrieval algorithm is therefore by analyzing the same dataset by two methods and looking for physical explanations of the potential cause of differences, as we have done in our manuscript.**

**We added the following discussion to the conclusions: "Further validation of the results presented here, e.g. with traverse measurements as in Salerno (2009a), would be advantageous. The authors of Salerno (2009a) found good agreement between their specific FRS evaluation for scanning spectrometers and car traverse measurements at Mt. Etna. However, comparing measurements from such different data sets is complex already for a single volcano, and additional error sources must be taken into account. The observation geometry needs to be controlled in a way that both instruments sample the same section of the plume and are subject to similar radiative transfer effects. The complexity of this can lead to an uncertainty, that can be observed in Fig. 6a of Salerno (2009a). Furthermore, the here studied volcanoes are not as easily accessible as Mt. Etna, and regularly conducted traverse measurements can not be obtained. A direct comparison of the differences observed at the two volcanoes would suffer from additional error**

sources due to the different set-up locations of the instruments relative to respective volcano, differences in local meteorological patterns (wind speeds and directions), terrain and subsequent dispersion patterns. "

Burton and Sawyer 2016 present an update to the Salerno et al. 2009a work in which they recognise that there are instrumental response features which add a fixed pattern to the measured spectra using CCD-based spectrometers, the 'flat' spectrum. This adds 1-3% noise but can be readily characterised and removed from each spectrometer using lab-based measurements of a broad-band UV source.

**Measuring a "flat spectrum" is not a new invention by Burton and Sawyer (2016). It is rather a standard technique used in many CCD detector applications, such as astrophotography, where it is called 'flat field correction'. The first source we found that mentions recordings of a halogen lamp to correct for pixel-to-pixel sensitivity in the context of DOAS measurements is Bussemer (1993), also the approach has been used in the GOME satellite spectrometers. Furthermore, these effects, and possibilities to correct for them are already discussed in the standard literature (Platt and Stutz, 2008). While characterizing these patterns in the laboratory certainly works it does not provide a solution for instruments that are inaccessible.**

The issues associated with the standard NOVAC analysis system have been well known for over a decade, since the implementation of the approach of Salerno et al. 2009a. The main objective of the Lubcke paper is that (page 3 line 1-3) "This work will follow the idea of using a high resolution Solar atlas spectrum (Chance and Kurucz, 2010) in order to calculate a gas free background spectrum which is used as an FRS for the DOAS evaluation of SO2.". It appears however that this work has already been done by Salerno et al. 2009, so the novelty of the current work should lie elsewhere.

**As discussed above, we see plenty of differences between our approach and Salerno et al. (2009) regarding implementation and applicability. Furthermore, we believe that a technical sound implementation of this important issue is necessary. Moreover, the quote from the manuscript mentions just one single part of the strategy we proposed to deal with the problem. There is much more original work in the article, including the implementation and feasibility of extension to a large operational network, the exploration of the circumstances under which the problem of 'SO$_2$ contamination' may arise and of the possible origins of the principal components of the residual structures.**

This immediately highlights that a major refocussing of the manuscript is needed, because so much of the work has already been developed. A valuable contribution of the Lubcke paper is the comparison of the SO2 offset correction with the Salerno et al. 2009a type retrieval for data from Nevada del Ruiz and Tungurahua. However, with 30 volcanoes in the NOVAC network and the fact that an established method for dealing with the issues of absolute SO2 retrievals has been applied for over a decade, it seems wholly inadequate that only two volcanoes are investigated. My recommendation is therefore to refocus this work from a rehash of previously published methods to a thorough investigation of the full implications of the application of the SO2 offset approach to the NOVAC network. Sincerely, with the time available since this issue was highlighted and a solution shown it is not acceptable to present just for two volcanoes from 30.

**While we agree that it is a certainly worthwhile goal of future work to investigate this issue at all 30 volcanoes, we must strongly reject the notion that this manuscript is a mere**

rehash of previous work. An in depth discussion of the retrieval approach is mandatory before applying such changes globally. Besides, as laid out above we are convinced that our contribution extends the scientific knowledge of spectroscopic evaluations. Furthermore, different from Salerno et al. (2009b) and Hibert et al. (2016) (who both only investigated one volcano) we added two further volcanoes with different measurement geometries. With the aim to improve the results from NOVAC, we have made a thorough analysis of how often and under which circumstances problems from contaminated reference spectra arise. We have then developed a method that can be implemented on a network basis.

A further requirement for a revised or resubmitted paper is that some comparison with traverse flux data is used. The final objective of all this work is to get as accurate as possible SO2 flux data, not SO2 slant column amounts. The corrections posed may have an angular dependence, leading to unexpected impacts on the reconstructed SO2 fluxes from scans. The only way to evaluate this properly is to compare scanner SO2 fluxes with traverse flux measurements, as performed by Salerno et al., 2009b, which should also be referenced here. I would therefore strongly recommend that such measurements and their comparison be included in a future revision, as otherwise the veracity of the final SO2 flux measurements will remain doubtful.

**As mentioned above, we will cite the work by Salerno et al, 2009b where appropriate (e.g. Page 2, Line 30 – 32 and in the conclusions). Regarding the traverse measurements, as the referee correctly points out, there are additional issues due to influences of the geometry and radiative transfer. Salerno et al., 2009b suggests "***that the accuracy of measurements obtained by the network is mostly dependent on the geometry of the plume, thus on the variability of the plume-transport direction***". We therefore believe, that the comparison with SO$_2$ traverse measurements would not greatly improve the understanding of the matter, in particular we believe that the request is unreasonable as it is not feasible to obtain traverse data at all 30 volcanoes. Not all volcanoes are as easily accessible as Mt. Etna. We suggest to use an FRS evaluation mainly to identify scans that are influenced by contaminated reference spectra and remove them from calculations of the average daily emission rate.**

The final contribution of the Lubcke paper is a PCA analysis which effectively does a qualitative job of removal of the flat spectrum as described by Burton & Sawyer 2016. The PCA approach means the user absolves themselves of responsibility for finding the physical process producing the observed features, which is rather disappointing, as precise physical understanding of the measurement process is how we can improve in the future. The majority of the features captured by PCA are flat spectrum, so this contribution is somewhat redundant, given that a clear physical explanation of the process is given by Burton & Sawyer 2016.

**We reject the reviewer's statement that '… is disappointing'. We think that we demonstrated in a large number of previous publications (e.g. Platt and Stutz 2008, Lübcke et al. 2013, Lampel et al. 2015 …) that we do care for and understand the underlying mechanisms influencing the performance of our instruments. The main reason for our careful interpretation of the principal components is that we cannot completely rule out further influences. However, as written in the manuscript, we agree with the referee and believe that the first component most likely corrects for instrumental effects (or as he/she calls it the "flat spectrum"). A constant difference of a trace gas between the Solar atlas**

**that we used and our measurement spectra could, just as an example, be a further influence on the principal components. We would also like to point out, that the precise physical understanding of the flat spectrum is not a novelty of Burton and Sawyer, 2016. This is rather a standard technique that is already discussed in Platt & Stutz, 2008. We did not apply the PCA for a lack of understanding or disregard of the instrumental effects but to obtain information from instruments that are otherwise inaccessible. To stress the point, in this way the methods allows for re-evaluation of historic data sets and to obtain coherent time series.**

**Furthermore, we have discussed the possible origins of the principal components and the conditions of appearance of the problem in working monitoring networks. There is nothing in the work cited by the referee that is more 'physical' than we had presented in ours. In both analyses the measured spectra are reconstructed from the adoption of the same physical picture of radiation entering the upper atmosphere and being scattered down into the instrument, and subjected to absorption by $O_3$ and $SO_2$ and by an approximation of the Ring effect, according to the extinction law. The broad components of molecular scattering are taken care of by inclusion of a polynomial in both methods, and only the characterization of the instrumental effects are dealt with in a different way in the two approaches (by laboratory characterization or by numerical analysis). Here lies the main disadvantage of Burton & Sawyer (2016) for our application, the need to measure spectra with a broad-band light source, preferably in the laboratory.**

**We adopted this simplified model because it is practical and works under several conditions for volcanic environments, but we do not claim that it is the most physically sound picture, as a detailed analysis of the physical process will require advanced radiative transfer modelling, e.g. by Monte Carlo simulations, which are as of yet impractical.**

The present work is also overly long with a redundant exhaustive explanation of the basic retrieval technique. Significant shortening is essential.

**We shortened the introduction and discussion of the basic retrieval technique as suggested by the referee.**

Literature:

Burton, M. R. and Sawyer, G. M.: iFit: An intensity-based retrieval for SO2 and BrO from scattered sunlight ultraviolet volcanic plume absorption spectra, Atmospheric Measurement Techniques Discussions, 2016, 1–47, doi:10.5194/amt-2015-380, http://www.atmos-meas-tech-discuss.net/amt-2015-380/, 2016.

Bussemer, M.: Der Ring-Effekt: Ursachen und Einfluß auf die spektroskopische Messung stratosphärischer Spurenstoffe, Diplomarbeit, University of Heidelberg, 1993.

Chance, K. and Kurucz, R. L.: An improved high-resolution solar reference spectrum for earth's atmosphere measurements in the ultraviolet, visible, and near infrared, Journal of Quantitative Spectroscopy and Radiative Transfer, 111, 1289–1295, doi:10.1016/j.jqsrt.2010.01.036, http://www.sciencedirect.com/science/article/pii/S0022407310000610, 2010.

Kurucz, R.L., Furenlid, I., Brault, J.,and Testerman, L., 1984. Solar flux atlas from to 1300 nm, Natl. Sol. Obs., Sunspot, New Mexico, 240.

Lampel, J., Frieß, U., and Platt, U.: The impact of vibrational Raman scattering of air on DOAS measurements of atmospheric trace gases, Atmos. Meas. Tech., 8, 3767-3787, doi:10.5194/amt-8-3767-2015, 2015

Lübcke, P., Bobrowski, N., Arellano, S., Galle, B., Garzón, G., Vogel, L., and Platt, U.: BrO/SO2 molar ratios from scanning DOAS measurements
in the NOVAC network, Solid Earth Discussions, 5, 1845–1870, doi:10.5194/sed-5-1845-2013, http://www.solid-earth-discuss.net/
5/1845/2013/, 2013.

Platt, U. and Stutz, J.: Differential Optical Absorption Spectroscopy - Principles and Applications, Physics of Earth and Space Environments,
35 Springer Berlin Heidelberg, 2008.

Salerno, G.G., Burton, M., Oppenheimer, C., Caltabiano, T., Tsanev, V.I., Bruno, N., 2009a, Novel retrieval of volcanic SO2 abundance from ultraviolet spectra. J. Volcanol. Geotherm. Res. 181, 141–153. doi:10.1016/j.jvolgeores.2009.01.009.

Salerno, G.G., Burton, M.R., Oppenheimer, C., Caltabiano, T., Randazzo, D., Bruno, N., Longo, V., 2009b, 183, 76-83. Three-years of SO2 flux measurements of Mt. Etna using an automated UV scanner array: comparison with conventional traverses and uncertainties in flux retrieval, doi:10.1016/j.jvolgeores.2009.02.013.